# TRANSFERRING HIERARCHICAL STRUCTURE WITH DUAL META IMITATION LEARNING

## ABSTRACT

Hierarchical Imitation learning (HIL) is an effective way for robots to learn sub-skills from long-horizon unsegmented demonstrations. However, the learned hierarchical structure lacks the mechanism to transfer across multi-tasks or to new tasks, which makes them have to learn from scratch when facing a new situation. Transferring and reorganizing modular sub-skills require fast adaptation ability of the whole hierarchical structure. In this work, we propose Dual Meta Imitation Learning (DMIL), a hierarchical meta imitation learning method where the high-level network and sub-skills are iteratively meta-learned with model-agnostic meta-learning. DMIL uses the likelihood of state-action pairs from each sub-skill as the supervision for the high-level network adaptation, and use the adapted high-level network to determine different data set for each sub-skill adaptation. We theoretically prove the convergence of the iterative training process of DMIL and establish the connection between DMIL and the Expectation-Maximization algorithm. Empirically, we achieve state-of-the-art few-shot imitation learning performance on the meta-world (Yu et al., 2019b) benchmark and comparable results on the Kitchen environment.

## 1 INTRODUCTION

Imitation learning (IL) has shown promising results for intelligent robots to conveniently acquire skills from expert demonstrations (Zhu et al., 2018; Peng et al., 2018). Nevertheless, acquiring skills from long-horizon unsegmented demonstrations has been a challenge for IL algorithms, because of the well-known issue of compounding errors (Ross et al., 2011). This is one of the crucial problems for the application of IL methods to robots since plenty of manipulation tasks are long-horizon. Hierarchical Imitation Learning (HIL) aims to tackle this problem by decomposing long-horizon tasks with a hierarchical model, in which a set of sub-skills are employed to accomplish specific parts of the long-horizon task, and a high-level network is responsible for determining the switching of sub-skills along with the task. Such a hierarchical structure is usually modeled with Options (Daniel et al., 2016; Krishnan et al., 2017; Jing et al., 2021) or goal-conditioned structure (Le et al., 2018). HIL expresses the nature of how humans solve complex tasks, and has been considered to be a valuable direction for IL algorithms (Osa et al., 2018).

However, most current HIL methods have no explicit mechanism to transfer previously learned skills to new tasks with few-shot demonstrations. This requirement comes from that the learned hierarchical structure may conflict with discrepant situations in new tasks. As shown in figure 1(a), both the high-level network and sub-skills need to be transferred to new forms to satisfy new requirements: the high-level network needs new manners to schedule sub-skills in new tasks (for example, calling different sub-skills at the same state), and each sub-skill needs to adapt to new specific forms in new tasks (for example, grasping different kinds of objects). Quickly adapting to new tasks can significantly increase the generalization ability of HIL methods and make them be applied to a wider range of scenarios, thus an appropriate approach needs to be exploited to endow HIL with such kind of ability with few-shot new task demonstrations.

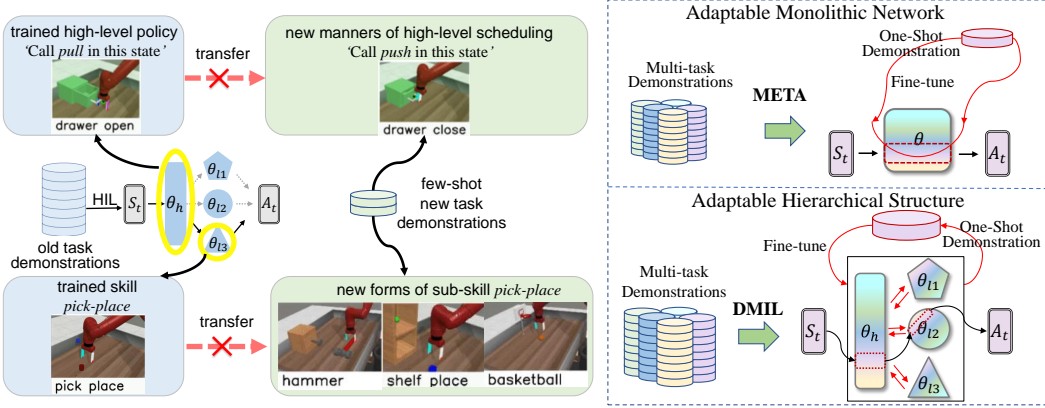

(a) Illustration of the transfer problem of HIL in new tasks.    (b) Comparison of MIL and DMIL.

Figure 1: (a) Both the high-level network and sub-skills need to be transferred to new tasks. Above: when the robot arm is over a half-open drawer, the task can be either opening or closing the drawer, which requires the high-level network to call different sub-skills. Below: a same sub-skill *pick-place* may exhibit different specific forms in new tasks. (b) DMIL aims to integrate MAML into HIL with a novel iterative optimization procedure that meta-learns both the high-level network and sub-skills.

Recently, meta imitation learning (MIL) (Yu et al., 2018b; Finn et al., 2017b; Yu et al., 2018a) employ model-agnostic meta-learning (MAML) (Finn et al., 2017a) to imitation learning procedure to enable the learned policy to quickly adapt to new tasks with few shot demonstrations. MAML first fine-tunes the policy network in the inner loop, then evaluates the fine-tuned network to update its initial parameters with end-to-end gradient descent at the outer loop. This inspires us to integrate MAML into HIL to transfer the hierarchical structure. However, most MIL methods meta-learn a monolithic policy in an end-to-end fashion, which does not conform to the iterative training of the bi-level structure in HIL, where the training supervision of each level comes from the other level. Integrating MAML into such a self-supervised bi-level optimization procedure is a novel problem. The main challenge comes from how to schedule the inner loops and outer loops of the meta-learning process of bi-level modules in HIL to ensure convergence.

In this work, we propose a novel hierarchical meta imitation learning framework called dual meta imitation learning (DMIL) to successfully incorporate MAML into the iterative training process of HIL, as shown in figure 1(b). We make the high-level network and sub-skills as mutual supervision during the bi-level MAML process: the likelihood of each state-action pair in few-shot demonstrations of each sub-skill can provide the supervisions for the meta-training of high-level network, and the fine-tuned high-level network in turn determines the data sets for the meta-training of each sub-skill, like a general EM process. We further design an elaborate training procedure for DMIL that first fine-tunes the high-level network and sub-skills *in sequence* at inner loops, then meta-updates them *simultaneously* at outer loops. We theoretically prove the convergence of this special training procedure by leveraging previous results from Ravi & Beatson (2019); Finn et al. (2018); Zou & Lu (2020) to reframe both MAML and DMIL as hierarchical Bayes inference processes and get the convergence of DMIL according to the convergence of MAML Fallah et al. (2020).

We test our method on the challenging meta-world benchmark environments (Yu et al., 2019b) and the Kitchen environment of D4RL benchmarks (Fu et al., 2020). In our experiments, we successfully acquire a set of meaningful sub-skills from a large scale of manipulation tasks, and achieve state-of-the-art few-shot imitation learning abilities in the ML45 suite. In summary, the main contributions of this paper are as follows:

- We propose DMIL, a novel hierarchical meta imitation learning framework that meta-learns both the high-level network and sub-skills from unsegmented multi-task demonstrations.
- We propose a special EM-like training algorithm to schedule the meta-learning and hierarchical imitation learning processes in DMIL and theoretically guarantee its convergence.
- We achieve state-of-the-art few-shot imitation learning performance on meta-world benchmark environments and comparable results in the Kitchen environment.

## 2 RELATED WORK

### 2.1 HIERARCHICAL IMITATION LEARNING

Recovering inherent sub-skills contained in expert demonstrations and then reuse them with hierarchical structures has long been an interesting topic in the hierarchical imitation learning (HIL) domain. According to whether there are pretraining tasks, we can divide HIL methods into two categories. The first kind aims to manually design a set of simple pretraining tasks that could encourage distinct skills or primitives, and then they learn a high-level network to master the switching of primitives to accomplish complex tasks (Florensa et al., 2017; Peng et al., 2019; Liu & Hodgins, 2017; Merel et al., 2019; Le et al., 2018). However, for unsegmented demonstrations where no pretraining tasks are provided, which is the situation in our paper, these methods can not be applied.

The second kind of methods aim to learn sub-skills with unsupervised learning methods. Daniel et al. (2016); Krishnan et al. (2017) acquire Options (Sutton et al., 1999) from demonstrations with an Expectation-Maximization-like procedure and use the Baum-Welch algorithm to estimate the parameters of different options. Henderson et al. (2018); Jing et al. (2021) integrate generative adversarial networks into option discovery process. Li et al. (2017); Sharma et al. (2019); Lee & Seo (2020) incorporate generative-adversarial imitation learning (Ho & Ermon, 2016) framework and an information-theoretic metric (Chen et al., 2016) to simultaneously imitate the expert and maximize the mutual-information between latent sub-skill categories and corresponding trajectories to acquire decoupled sub-skills. There are some methods called mixture-of-expert (MoE) that compute the weighted sum of all primitives to get the action rather than only using one of them at each time step (Hausknecht & Stone, 2016; Neumann et al., 2009; Jacobs et al., 1991). Other methods aim to seek an appropriate latent space that can map sub-skills into it and then condition a policy on the latent variable to reuse sub-skills (Lynch et al., 2019; Haarnoja et al., 2018; Hausman et al., 2018; Hakhamaneshi et al., 2021; Price & Boutilier, 2003).

For transferring learned sub-skills, some work fine-tune the whole structure in new tasks (Hakhamaneshi et al., 2021). However, the performance of fine-tuning all depends on the generalization of deep networks, which may vary among different tasks and network designs.

### 2.2 META IMITATION LEARNING

Meta imitation learning, or one-shot imitation learning, leverages various meta-learning methods and multi-task demonstrations to meta-learn a policy that can be quickly adapted to a new task with few-shot new task demonstrations. Duan et al. (2017); Cachet et al. (2020) employ self-attention modules to process the whole demonstration and the current observation to predict current action. Yu et al. (2018a); Finn et al. (2017b); Yu et al. (2018b) use model-agnostic meta-learning (MAML) (Finn et al., 2017a) to achieve one-shot imitation learning ability for various manipulation tasks with robot or human visual demonstrations. Xu et al. (2019); Yu et al. (2019a) propose to meta-learn a robust reward function that can be quickly adapted to new tasks and then use it to perform IRL in new tasks. However, they need downstream inverse reinforcement learning after the adaptation of reward functions, thus conflicts with our goal of few-shot adaptation. Most of above methods only learn one monolithic policy, lacking the ability to model multiple sub-skills in long-horizon tasks. Some works aim to tackle the multi-modal data problem in meta-learning by avoiding single parameters initialization across all tasks (Vuorio et al., 2019; Alet et al., 2018; Frans et al., 2018; Yao et al., 2019), but they lack the mechanism to schedule the switching of different sub-skills over time. There are some works that also meta-learn a set of sub-skills in a hierarchical structure (Yu et al., 2018a; Frans et al., 2018), but they either use manually designed pretraining tasks or relearn the high-level network in new tasks, which is not appropriate in few-shot imitation learning settings.

## 3 METHOD

### 3.1 PRELIMINARIES

We denote a discrete-time finite-horizon Markov decision process (MDP) as a tuple $(\mathcal{S}, \mathcal{A}, T, P, r, \rho_0)$, where $\mathcal{S}$ is the state space, $\mathcal{A}$ is the action space, $T$ is the time horizon,

Figure 2: The iterative meta-learning process of DMIL at each iteration. Left: the supervision of high-level network (sub-skill categories) comes from the most accurate sub-skill (the green one, sub-skill 1 here). Right: the sub-skill updated at current step (the green one, sub-skill 0 here) is determined by the fine-tuned high-level network.

$P : \mathcal{S} \times \mathcal{A} \times \mathcal{S} \rightarrow [0, 1]$ is the transition probability distribution, $r : \mathcal{S} \times \mathcal{A} \rightarrow \mathbb{R}$ is the reward function, and $\rho_0$ is the distribution of the initial state $s_0$.

## 3.2 FORMULATION OF META IMITATION LEARNING PROBLEM

We firstly introduce the general setting of the meta imitation learning problem. The goal of meta imitation learning is to extract some common knowledge from a set of robot manipulation tasks $\{\mathcal{T}_i\}$ that come from the same task distribution $p(\mathcal{T})$, and adapt it to new tasks quickly with few shot new task demonstrations. As in model-agnostic meta-learning algorithm (MAML) (Finn et al., 2017a), we formalize the common knowledge as the initial parameter $\theta$ of the policy network $\pi_\theta$ that can be efficiently adapted with new task gradients.

For each task $\mathcal{T}_i \sim p(\mathcal{T})$, a set of demonstrations $\mathcal{D}_i$ is provided, where $\mathcal{D}_i$ consists of $N$ demonstration trajectories: $\mathcal{D}_i = \{\tau_{ij}\}_{j=1}^N$, and $\tau_{ij}$ consists of a sequence of state-action pairs: $\tau_{ij} = \{(s_t, a_t)\}_{t=1}^{T_{ij}}$, where $T_{ij}$ is the length of $\tau_{ij}$. Each $\mathcal{D}_i$ is randomly split into support set $\mathcal{D}_i^{tr}$ and query set $\mathcal{D}_i^{val}$ for meta-training and meta-testing respectively. During the training phase, we sample $m$ tasks from $p(\mathcal{T})$, and in each task $\mathcal{T}_i$, we use $\mathcal{D}_i^{tr}$ to fine-tune $\pi_\theta$ to get the adapted task-specific parameters $\lambda_i$ with gradient descent, and then evaluate it with $\mathcal{D}_i^{val}$ to get the meta-gradient of $\mathcal{T}_i$, and we optimize the initial parameters $\theta$ with the average of meta-gradients from all $m$ tasks. As in Finn et al. (2017b); Duan et al. (2017), we use behavior cloning (Atkeson & Schaal, 1997) loss as our metrics for meta-training and meta-testing. It aims to train a policy $\pi_\theta : \mathcal{S} \rightarrow \mathcal{A}$ that maximizes the likelihood such that $\theta^* = \arg\max_\theta \sum_{i=1}^N \log \pi_\theta(a_i|s_i)$, where $N$ is the number of provided state-action pairs. We denote the loss function of this optimization problem as $\mathcal{L}_{BC}(\theta, \mathcal{D})$, and the general objective of meta imitation learning problem is:

$$\min_\theta \sum_{i=1}^m \mathcal{L}_{BC}\left(\lambda_i, \mathcal{D}_i^{\text{val}}\right), \tag{1}$$

where $\lambda_i = \theta - \alpha \nabla_\theta \mathcal{L}_{BC}(\theta, \mathcal{D}_i^{tr})$, and $\alpha$ is a hyper-parameter which represents the inner-update learning rate.

## 3.3 DUAL META IMITATION LEARNING (DMIL)

In this work we assume at each time step $t$, the robot may switch to different sub-skills to accomplish the task. We define the sub-skill category at each time step $t$ as $z_t = 1, \cdots, K$, where $K$ is the maximum number of sub-skills. We assume a successful trajectory $\tau_{ij}$ of a task $\mathcal{T}_i$ is generated from several (at least one) sub-skill policies, i.e., $\tau_{ij} = \sum_{t=1}^{T_{ij}}\{(s_t, \pi_E(s_t|z_t))\}$, where $\pi_E$ represents the expert policy. Our goal is to learn a hierarchical structure from multi-task demonstrations $\{\mathcal{D}_1, \cdots, \mathcal{D}_m\}$ in an unsupervised fashion. In our model, a high-level network $\pi_{\theta_h}$ that parameterized by $\theta_h$ determines the sub-skill category $\hat{z}_t$ at each time step $t$, and the $z$-th sub-skill among $K$ different sub-skills $\pi_{\theta_{l1}}, \cdots, \pi_{\theta_{lK}}$ will be called to predict the corresponding action $\hat{a}_t$ of state $s_t$, where the hat symbol denotes the predicted result. We use $\lambda_h$ and $\lambda_{l1}, \cdots, \lambda_{lk}$ to represent the adapted parameters of $\theta_h$ and $\theta_{l1}, \cdots, \theta_{lK}$ respectively. In order to achieve few-shot learning ability in new tasks, we condition the high-level network only on states, i.e., $\hat{z}_t = \pi_{\theta_h}(s_t)$, since at the testing phase we only have access to states and have no access to action information.

DMIL aims to first fine-tune both $\pi_{\theta_h}$ and $\pi_{\theta_{l0}}, \cdots, \pi_{\theta_{lK}}$ and then meta-update them. In a new task, $\pi_{\theta_h}$ may not provide correct sub-skill categories as stated in the introduction. However, sub-skills still retain the ability to give out supervision for the high-level network with knowledge learned from previously learned tasks and few-shot demonstrations. This is because most robot manipulation tasks are made up of a set of shared basis skills like *reach, push* and *pick-place*. As shown in the left side of 2, the sub-skill that gives out the closet $\hat{a}_t$ to $a_t$ can be seen as supervision for the high-level network to classify $s_t$ into this sub-skill. On the other hand, the adapted high-level network can classify each data point in provided demonstrations to different sub-skills for them to perform fine-tuning, as shown in the right side of figure 2. In summary, DMIL contains four steps for one training iteration. We call them **High-Inner-Update (HI)**, **Low-Inner-Update (LI)**, **High-Outer-Update (HO)**, and **Low-Outer-Update (LO)**, which represents the fine-tuning and meta-updating process of the bi-level networks respectively. The key problem is how to arrange these optimization orders to ensure convergence. We first introduce these four steps formally here, then discuss how to schedule them in the next section. The whole procedure is summarized in algorithm 1.

**HI:** For each sampled task $\mathcal{T}_i$, we sample the first batch of trajectories $\{\tau_{i1}\}$ from $\mathcal{D}_i^{tr}$. The principle of this step is to use sub-skill that can predict the closest action to the expert action to provide self-supervised category ground truths for the training of high-level network, which is a classifier in form. We make *every* state-action pair passed directly to each sub-skill and compute $\mathcal{L}_{BC}(\theta_{lk}, \tau_{i1}), k = 1, \cdots, K$, and choose the ground truth at each time step as the sub-skill category $k$ that minimizes $\mathcal{L}_{BC}(\theta_{lk}, (s_t, a_t))$ :

$$p(z_{i1t} = k) = \begin{cases} 1, & \text{if } k = \arg\min_k \mathcal{L}_{BC}(\theta_{lk}, (s_t, a_t)) \\ 0, & \text{else} \end{cases} . \tag{2}$$

Then we get predicted sub-skill categories from the high-level network: $\hat{z_{i1t}} = \pi_{\theta_h}(s_t)$, and use a cross-entropy loss to train the high-level network:

$$\mathcal{L}_h(\theta_h, \tau_{i1}) = -\frac{1}{T_{i1}} \sum_{t=1}^{T_{i1}} \sum_{k=1}^{K} p(z_{i1t} = k) \log p(\hat{z_{i1t}} = k). \tag{3}$$

Finally we perform gradient descent on the high-level network and get $\lambda_h = \theta_h - \alpha \nabla_{\theta_h} \mathcal{L}_h(\theta_h, \tau_{i1})$. Note $\theta_{l1}, \cdots, \theta_{lk}$ are freezed here.

**LI:** We sample the second batch of trajectories $\{\tau_{i2}\}$ from $\mathcal{D}_i^{tr}$. The adapted high level network $\pi_{\lambda_h}$ will process each state in $\tau_{i2}$ to get sub-skill category $\hat{z_{i2t}} = \pi_{\lambda_h}(s_t)$ at each time step, thus we get $K$ separate data sets for different sub-skills: $\mathcal{D}_{2k} = \{(s_{i2t}, a_{i2t}) | \hat{z_{i2t}} = k\}, k = 1, \cdots, K$. Then we compute the adaptation loss for each sub-skill with the corresponding dataset. In case of continuous action space, we assume that actions belong to Gaussian distributions, so we have:

$$\mathcal{L}_{BC}(\theta_{lk}, \mathcal{D}_{2k}) = -\frac{1}{N_k} \sum_{t=1}^{N_k} (a_t - \pi_{\theta_{lk}}(s_t))^2, \tag{4}$$

where $N_k$ is the number of state-action pairs in $\mathcal{D}_{2k}$. Finally we perform gradient descent on sub-skills and get $\lambda_{lk} = \theta_{lk} - \alpha \nabla_{\theta_{lk}} \mathcal{L}_{BC}(\theta_{lk}, \mathcal{D}_{2k})$. Note $\pi_{\lambda_h}$ is frozen in this process.

**HO:** We sample the third batch of trajectories $\{\tau_{i3}\}$ from $\mathcal{D}_i^{tr}$ and get $\mathcal{L}(\lambda_h, \tau_{i3})$ as in the HI process. Then we compute meta-gradient for $\theta_h$ with $\mathcal{L}(\lambda_h, \tau_{i3})$ as follows:

$$\nabla_{\theta_h} \mathcal{L}(\lambda_h, \tau_{i3}) = \nabla_{\lambda_h} \mathcal{L}(\lambda_h, \tau_{i3})|_{\lambda_h = \theta_h - \alpha \nabla_{\theta_h} \mathcal{L}(\theta_h, \tau_{i2})} * \nabla_{\theta_h} \lambda_h. \tag{5}$$

**LO:** we sample $\tau_{i4}$ and get $\mathcal{L}(\lambda_{lk}, \mathcal{D}_{4k}), k = 1, \cdots, K$ as in the LI process, then we compute meta-gradient for $\theta_{lk}$ with $\mathcal{L}(\lambda_{lk}, \mathcal{D}_{4k})$ as follows:

$$\nabla_{\theta_{lk}} \mathcal{L}(\lambda_{lk}, \mathcal{D}_{4k}) = \nabla_{\lambda_{lk}} \mathcal{L}(\lambda_{lk}, \mathcal{D}_{4k})|_{\lambda_{lk} = \theta_{lk} - \alpha \nabla_{\theta_{lk}} \mathcal{L}(\theta_{lk}, \mathcal{D}_{1k})} * \nabla_{\theta_{lk}} \lambda_{lk}. \tag{6}$$

Note after the training of $m$ tasks, we average all meta-gradients from $m$ tasks and perform gradient descents on the initial parameters *together* to update high-level parameters $\theta_h' = \theta_h - \beta \sum_{i=1}^m \nabla_{\theta_h} \mathcal{L}(\lambda_h, \tau_{i3})$ and sub-skill policies parameters $\theta_{lk}' = \theta_{lk} - \beta \sum_{i=1}^m \nabla_{\theta_{lk}} \mathcal{L}(\lambda_{lk}, \tau_{i4})$, $k = 1, \cdots, K$, i.e., we do not update them at step 5 and 6. This is crucial to ensure convergence.

For testing, although our method needs totally two batches of trajectories for one round of adaptation, in practice we find only using one trajectory to perform HI and LI also works well in new tasks, thus DMIL can satisfy the one-shot imitation learning requirement. Besides the above process, we also add an auxiliary loss to better drive out meaningful sub-skills to avoid the excessively frequently switching between different sub-skills along with time. Detailed information can be found in B.

## 4 THEORETICAL ANALYSIS

DMIL is a novel iterative hierarchical meta-learning procedure, and its convergence of DMIL needs to be proved to ensure feasibility. As stated in the above section, what makes DMIL special is that in **HI** and **LI**, we update parameters of each module immediately, but in **LO** and **HO**, we store the gradients of each part and update them simultaneously. In this section, we show this can make DMIL converge by rewriting both MAML and DMIL as hierarchical variational Bayes problems to establish the equivalence between them, since the convergence of MAML can be proved in Fallah et al. (2020). Proofs of all theorems are in Appendix C.

### 4.1 HIERARCHICAL VARIATIONAL BAYES FORMULATION OF MAML

According to (Ravi & Beatson, 2019), MAML is a hierarchical variational Bayes inference process. The general meta-learning objective 1 can be formulated as follows:

$$\mathcal{L}(\theta, \lambda_1, \cdots, \lambda_m) = \log \left[ \prod_{i=1}^{m} p\left(\mathcal{D}_i | \theta\right) \right] \geq \sum_{i=1}^{m} \{ \mathrm{KL}(q(\phi_i; \lambda_i) \| p(\phi_i | \mathcal{D}_i, \theta)) \\ + E_{q(\phi_i; \lambda_i)}[\log p(\mathcal{D}_i, \phi_i | \theta) - \log q(\phi_i; \lambda_i)] \}, \tag{7}$$

where $\phi_i, i = 1, \cdots, m$ represent the local latent variables for task $\mathcal{T}_i$, and $\lambda_1, \cdots, \lambda_M$ are the variational parameters of the approximate posteriors over $\phi_1, \cdots, \phi_M$. We denote $\lambda_i$ as $\lambda_i(\mathcal{D}_i, \theta)$ and $p(\phi_i | \mathcal{D}_i, \theta)$ as $p(\phi_i | \mathcal{D}_i^{tr}, \theta)$ to mean that $\lambda_i$ and $\phi_i$ are determined with prior parameters $\theta$ and support data $\mathcal{D}_i^{tr}$. First we need to minimize $\mathrm{KL}(q(\phi_i; \lambda_i) \| p(\phi_i | \mathcal{D}_i^{tr}, \theta))$ w.r.t. $\lambda_i$. According to C.2, we have:

$$\lambda_i(\mathcal{D}_i^{tr}, \theta) = \arg\max_{\lambda_i} E_{q(\phi_i; \lambda_i)}[\log p(\mathcal{D}_i^{tr} | \phi_i)] - \mathrm{KL}(q(\phi_i; \lambda_i) \| p(\phi_i | \theta)), \tag{8}$$

and we can establish the connection between 8 and the fine-tuning process in MAML by the following Lemma:

**Lemma 1** In case $q(\phi_i; \lambda_i)$ is a Dirac-delta function and choosing Gaussian prior for $p(\phi_i | \theta)$, equation 8 equals to the inner-update step of MAML, that is, maximizing $\log p(\mathcal{D}_i^{tr})$ w.r.t. $\lambda_i$ by early-stopping gradient-ascent with choosing $\mu_\theta$ as initial point:

$$\lambda_i(\mathcal{D}_i^{tr}; \theta) = \mu_\theta + \alpha \nabla_\theta \log p\left(\mathcal{D}_i^{tr} | \theta\right)|_{\theta=\mu_\theta}. \tag{9}$$

Then we need to optimize $\mathcal{L}(\theta, \lambda_1, \cdots, \lambda_M)$ w.r.t. $\theta$. Since we evaluate $p(\mathcal{D}_i | \lambda_i(\mathcal{D}_i^{tr}, \theta))$ with only $\mathcal{D}_i^{val}$, we assume $p(\mathcal{D}_i | \lambda_i(\mathcal{D}_i^{tr}, \theta)) = p(\mathcal{D}_i^{val} | \lambda_i(\mathcal{D}_i^{tr}, \theta))$. We give out the following theorem to establish the connection between the meta-update process and the optimization of $\mathcal{L}(\theta, \lambda_1, \cdots, \lambda_M)$:

**Theorem 1** In case that $\Sigma_\theta \to 0^+$, i.e., the uncertainty in the global latent variables $\theta$ is small, the following equation holds:

$$\nabla_\theta \mathcal{L}(\theta, \lambda_1, \cdots, \lambda_M) = \sum_{i=1}^{M} \nabla_{\lambda_i} \log p(\mathcal{D}_i^{val} | \lambda_i) \cdot \nabla_\theta \lambda_i(\mathcal{D}_i^{tr}, \theta). \tag{10}$$

A general EM algorithm will first compute the distribution of latent variables (E-step), then optimize the joint distribution of latent variable and trainable parameters (M-step), and the likelihood of data can be proved to be monotone increasing to guarantee the convergence since the evidence lower bound of likelihood is monotone increasing. Here $\phi_i, i = 1, \cdots, M$ are the latent variables, and $\theta$ corresponds to the trainable parameters. Lemma 1 and Theorem 1 correspond to the E-step and M-step respectively. In the following part we establish the equivalence between 9 with 3 and 4, and between 10 with 5 and 6 to prove the equivalence between DMIL and MAML.

### 4.2 MODELING DMIL WITH HIERARCHICAL VARIATIONAL BAYES FRAMEWORK

For simplicity, here we only derive in one specific task $\mathcal{T}_i$, since derivatives of parameters from multi-task can directly add up. We first establish the connection between the maximization of $\log p(\mathcal{D}_i^{tr} | \theta_h, \theta_{l1}, \cdots, \theta_{lK})$ with the particular loss functions in DMIL:

**Theorem 2** In case of $p(a_t|s_t, \theta_{lk}) \sim \mathcal{N}(\mu_{\theta_{lk}(s_t)}, \sigma^2)$, we have:

$$\nabla_{\theta_h} \log p(\mathcal{D}_i^{tr}|\theta_h, \theta_{l1}, \cdots, \theta_{lK}) = \nabla_{\theta_h} \mathcal{L}_h(\theta_h, \mathcal{D}_i^{tr}), \tag{11}$$

and

$$\nabla_{\theta_{lk}} \log p(\mathcal{D}_i^{tr}|\theta_h, \theta_{l1}, \cdots, \theta_{lK}) = \nabla_{\theta_{lk}} \mathcal{L}_{BC}(\theta_{lk}, \mathcal{D}_{2k}), k = 1, \cdots, K. \tag{12}$$

Note in 12, $\mathcal{D}_{2k}$ corresponds to data sets determined by the adapted high level network $\lambda_h$, and this connects with 3 and 4 in DMIL. According to 8, finding $\lambda_i$ equals to maximize $\log p(\mathcal{D}_i^{tr}|\theta)$ in specific conditions, and here in Theorem 2, we prove that maximize $\log p(\mathcal{D}_i^{tr}|\theta_h, \theta_{l1}, \cdots, \theta_{lK})$ corresponds to 3 and 4 in DMIL. Thus theorem 2 corresponds to the E-step of DMIL, where we take $\tau_{i1}$ and $\tau_{i2}$ as $\mathcal{D}_i^{tr}$, and optimize $\arg\max_{\lambda_i} E_{q(\phi_i;\lambda_i)}[\log p(\mathcal{D}_i^{tr}|\phi_i)] - \mathrm{KL}(q(\phi_i;\lambda_i)\|p(\phi_i|\theta))$ with coordinate descent method, which can be proved to be equal to 9 in C.5.

For the M-step, we take $\tau_{i3}$ and $\tau_{i4}$ as $\mathcal{D}_i^{val}$. According to Theorem 1, we can take the derivative of $\lambda_{ih}, \lambda_{il1}, \cdots, \lambda_{ilK}$ to maximize the joint distribution of latent variables and trainable parameters to maximize the likelihood of dataset, so we have:

$$\nabla_{\theta_h, \theta_l} \log p(\mathcal{D}_i^{val}|\lambda_{ih}, \lambda_{il}) = [\nabla_{\lambda_{ih}} \log p(\mathcal{D}_i^{val}|\lambda_{ih}) * \nabla_{\theta_h} \lambda_{ih}(\mathcal{D}_i^{tr}, \theta_h),$$
$$\nabla_{\lambda_{il}} \log p(\mathcal{D}_i^{val}|\lambda_{il}) * \nabla_{\theta_l} \lambda_{il}(\mathcal{D}_i^{tr}, \theta_l)]^T \tag{13}$$

where $\theta_{il} = [\theta_{i1}, \cdots, \theta_{iK}]^T$ and $\lambda_{il} = [\lambda_{i1}, \cdots, \lambda_{iK}]^T$. This is exactly the gradients computed in HO and LO steps. Note this computation process can be automatically accomplished with standard deep learning libraries such as PyTorch (Paszke et al., 2019). To this end, we establish the equivalence between DMIL and MAML, and the convergence of DMIL can be proved.

For a clearer comparison, MAML is an iterative process of $\theta \to \lambda \to \theta'$, and DMIL is an iterative process of $\theta_h, \theta_l \to \lambda_h, \theta_l \to \lambda_h, \lambda_l \to \theta'_h, \theta'_l$, where the posterior estimation stages $\theta_h, \theta_l \to \lambda_h, \theta_l \to \lambda_h, \lambda_l$ has no effect on parameters $\theta_h, \theta_l$, thus can be divided to two steps as in DMIL. This decoupled fine-tuning fashion is exactly what we need to first adapt the high-level network and then adapt sub-skills. If we end-to-end fine-tune parameters like $\theta_h, \theta_l \to \lambda_h, \lambda_l$, sub-skills will receive supervisions from an unadapted high-level network, which may provide incorrect classifications. Different to this, the meta-updating process $\lambda_h, \lambda_l \to \theta'_h, \theta'_l$ must be done at the same time, since if we update $\theta_h$ and $\theta_l$ successively, the later one will receive different derivative (for example, $\nabla_{\theta_l} \log p(\mathcal{D}_i^{val}|\theta'_{ih}, \lambda_{il})$ ) from derivatives in MAML ($\nabla_{\theta_l} \log p(\mathcal{D}_i^{val}|\lambda_{ih}, \lambda_{il})$), and the equivalence would not be proved.

## 5 EXPERIMENTS

In experiments we aim to answer the following questions: (a) Can DMIL successfully transfer the learned hierarchical structure to new tasks with few-shot new task demonstrations? (b) Can DMIL achieve higher performance compared to other few-shot imitation learning methods? (c) What are effects of different parts in DMIL, such as the skill number $K$, the bi-level meta-learning procedure, and the continuity constraint? Video results are provided in supplementary materials.

### 5.1 ENVIRONMENTS AND EXPERIMENT SETUPS

We choose to evaluate DMIL on two representative robot manipulation environments. The first one is Meta-world benchmark environments (Yu et al., 2019b), which contains 50 diverse robot manipulation tasks, as shown in figure 6 and 7. We use both ML10 suite (10 meta-training tasks and 5 meta-testing tasks) and ML45 suite (45 meta-training tasks and 5 meta-testing tasks) to evaluate our method, and collect 2K demonstrations for each task. We choose Meta-world since we think a large scale of diverse manipulation tasks can help our method to learn semantic skills. We use the following approaches for comparison in this environment: **Option-GAIL**: a hierarchical generative adversarial imitation learning method to discover options from unsegmented demonstrations (Jing et al., 2021). We use Option-GAIL to evaluate the effect of meta-learning in DMIL. **MIL**: a transformer-based meta imitation learning method (Cachet et al., 2020). We use MIL to evaluate the effect of the hierarchical structure in DMIL. **MLSH**: the meta-learning shared hierarchies method (Frans et al., 2018) that relearns the high-level network in every new task. We use MLSH to evaluate the effect of fine-tune (rather than relearn) the high-level network in new tasks. **PEMIRL**: a contextual meta inverse RL method which transfers the reward function in the new tasks (Yu et al., 2019a). We use PEMIRL to show DMIL can transfer to new tasks that have significantly different reward functions.

Table 1: Success rates of different methods on Meta-world environments with $K = 3$. Each data point comes from the success rate of 20 tests.

| | ML10 | | | | ML45 | | | |
| | Meta-training | | Meta-testing | | Meta-training | | Meta-testing | |
| Methods | 1-shot | 3-shot | 1-shot | 3-shot | 1-shot | 3-shot | 1-shot | 3-shot |
|---|---|---|---|---|---|---|---|---|
| OptionGAIL | 0.455±0.011 | **0.952±0.016** | 0.241±0.042 | 0.640±0.025 | 0.506±0.008 | 0.715±0.006 | 0.220±0.013 | 0.481±0.010 |
| MIL | **0.776±0.025** | 0.869±0.029 | 0.361±0.040 | 0.689±0.032 | 0.584±0.011 | 0.745±0.017 | 0.205±0.024 | 0.510±0.005 |
| PEMIRL | 0.598±0.023 | 0.810±0.007 | 0.162±0.003 | 0.256±0.009 | 0.289±0.051 | 0.396±0.024 | 0.105±0.005 | 0.126±0.008 |
| MLSH | 0.506±0.134 | 0.725±0.021 | 0.106±0.032 | 0.135±0.009 | 0.235±0.093 | 0.295±0.021 | 0.050±0.000 | 0.050±0.000 |
| DMIL | 0.775±0.010 | 0.949±0.009 | **0.396±0.016** | **0.710±0.021** | **0.590±0.010** | **0.859±0.008** | **0.376±0.004** | **0.640±0.009** |

Table 2: Rewards of different methods on four unseen tasks in Kitchen environment with $K = 4$. Boldface indicates excluded objects during training.

| Task (Unseen) | FIST-no-FT | SPiRL | DMIL(ours) |
|---|---|---|---|
| Microwave, Kettle, **Top Burner**, Light Switch | 2.0 ± 0.0 | **2.1 ± 0.48** | 1.5±0.48 |
| **Microwave**, Bottom Burner, Light Switch, Slide Cabinet | 0.0 ± 0.0 | 2.3 ± 0.49 | **2.35±0.39** |
| Microwave, **Kettle**, Hinge Cabinet, Slide Cabinet | 1.0 ± 0.0 | 1.9 ± 0.29 | **3.15±0.22** |
| Microwave, Kettle, Hinge Cabinet, **Slide Cabinet** | 2.0 ± 0.0 | **3.3 ± 0.38** | 2.95±0.44 |

The second one is the Kitchen environment of the D4RL benchmark (Fu et al., 2020), which contains five different sub-tasks in the same kitchen environment. The accomplishment of each episode requires sequentially completions of four specific sub-tasks, as shown in figure 9. We use an open demonstration data set (Gupta et al., 2019) to train our method. During training, we exclude interactions with selected objects and at test-time we provide demonstrations that involve manipulating the excluded object to make them as unseen tasks. We choose this environment to show DMIL can be used in long-horizon tasks and is robust across different environments. We use two approaches for comparison in this experiment: **SPiRL**: an extension of the skill extraction methods to imitation learning over skill space (Price & Boutilier, 2003); **FIST**: an algorithm that extracts skills from offline data with an inverse dynamics model and a distance function (Hakhamaneshi et al., 2021).

We use fully-connected neuron networks for both the high-level network and sub-skills. Detailed descriptions on the environment setup, demonstration collection procedure, hyper-parameters setting, training process, and descriptions of different methods can be found in appendix E.

## 5.2 RESULTS

Table 1 shows success rates of different methods in ML10 and ML45 suites with sub-skill number $K = 3$. We perform 1-shot and 3-shot experiments respectively to show the progressive few-shot performance of different methods. DMIL achieves the best results in ML10 testing suite and ML45 training and testing suites. This shows the superiority of our method for transferring across a large scale of manipulation tasks. OptionGAIL achieves high success rates in both ML10 and ML45 training suites. These results come from their hierarchical structures that have adequate capacity to fit potential multi-modal behaviors in multi-task demonstrations. MIL achieves comparable results for all meta-testing tasks but is worse than DMIL. This shows the necessity of meta-learning processes. Compared to them, PEMIRL and MLSH are mediocre among all suites. This comes from that the reward functions across different tasks are difficult to transfer with few shot demonstrations, and the relearned high-level network of MLSH damages previously learned knowledge. We also illustrate the t-sne results of these methods in figure 4(a) to further analyze the methods in appendix D.2.

Table 2 shows the rewards of different methods on four unseen tasks in the Kitchen environment. *FIST-no-FT* refers to a variant of FIST that does not use future conditioning, which makes the comparison fairer. DMIL achieves higher rewards on two out of four tasks and comparable results on the other two tasks, which exhibits the effectiveness of the bi-level meta-training procedure. The poor performance of DMIL on the first task may come from the choice of skill number $K$ or from low-quality demonstrations. We perform ablation studies on $K$ in the next section.

Figure 3 shows curves of sub-skill probabilities along time of two tasks *window-close* and *peg-insert-side* of Meta-world and the *microwave-kettle-top burner-light* task in Kitchen environment. We can see the activation of sub-skills shows a strong correlation to different stages of tasks. In first two tasks, $\pi_{\theta_{l_0}}$ activates when the robot is closing to something, $\pi_{\theta_{l_1}}$ activates when the robot is picking up something, and $\pi_{\theta_{l_2}}$ activates when the robot is manipulating something. In the third task, $\pi_{\theta_{l_2}}$ activates when the robot is manipulating the microwave, $\pi_{\theta_{l_0}}$ activates when the robot is

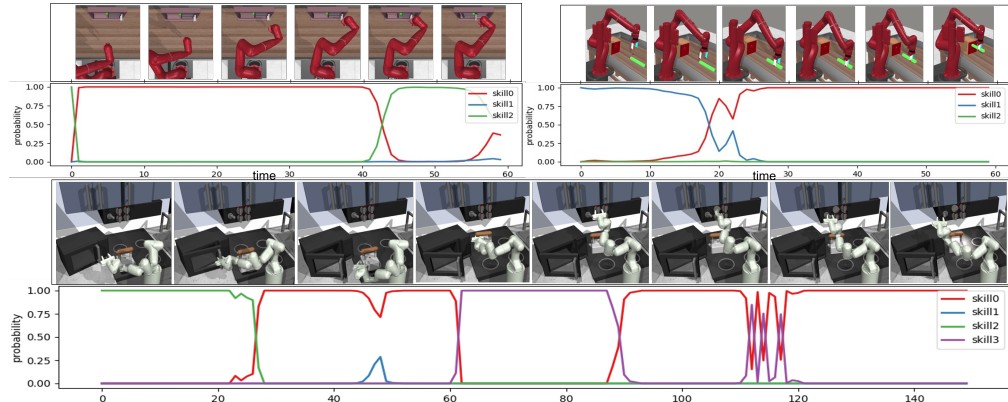

Figure 3: The iterative meta-learning process of DMIL at each iteration. Left: the supervision of high-level network (sub-skill category) comes from the most accurate sub-skill. Right: the sub-skill updated at current step is determined by the fine-tuned high-level network.

manipulating the kettle or the light switch, and $\pi_{\theta_{l3}}$ activates when the robot is manipulating the burner switch. This shows that DMIL has the ability to learn semantic sub-skills from unsegmented multi-task demonstrations.

## 5.3 ABLATION STUDIES

In this section we perform ablation studies on different components of DMIL to provide effects of different parts. Due to limited space, we put ablations on fine-tuning steps, bi-level meta-learning processes, continuity constraint and hard/soft EM choices in appendix D.

Table 3: Success rates of different sub-skill number in Meta-world environments.

| | ML10 | | | | ML45 | | | |
|---|---|---|---|---|---|---|---|---|
| | Meta-training | | Meta-testing | | Meta-training | | Meta-testing | |
| K | 1-shot | 3-shot | 1-shot | 3-shot | 1-shot | 3-shot | 1-shot | 3-shot |
| 2 | 0.76 | 0.955 | 0.32 | **0.72** | 0.563 | 0.818 | **0.44** | **0.67** |
| 3 | 0.775 | 0.949 | 0.396 | 0.71 | 0.59 | 0.859 | 0.376 | 0.64 |
| 5 | 0.795 | 0.94 | **0.52** | 0.57 | 0.713 | 0.92 | 0.21 | 0.48 |
| 10 | **0.8** | **0.975** | 0.38 | 0.62 | **0.736** | **0.931** | 0.34 | 0.64 |

**Effect of different skill number** $K$: Table 3 shows the effect of different sub-skill number $K$ in Meta-world experiments. We can see that a larger $K$ can lead to higher success rates on meta-training tasks, but a smaller $K$ can lead to better results on meta-testing tasks. This tells us that an excessive number of sub-skills may result in over-fitting on training data, and a smaller $K$ can play the role of regularization. In Kitchen experiments, we can see similar phenomenons in table 5. It is worth noting in both environments, we did not encounter collapse problems, i.e., every sub-skill gets well-trained even when $K = 8$ in kitchen environment or $K = 10$ in Meta-world environments. This may come from that, in the meta-training stage, adding more sub-skills can help the whole structure get lower loss, thus DMIL will use all of them for training. However, in our supplementary videos, we can see sub-skills trained with a large $K$ (for instance, $K = 10$ in Meta-world environments) are not as semantic as sub-skills trained by a small $K$ (for instance, $K = 3$ in Meta-world environments) during the execution of a task.

## 6 DISCUSSION AND FUTURE WORKS

In this work, we propose DMIL to meta-learn a hierarchical structure from unsegmented multi-task demonstrations to endow it with fast adaptation ability to transfer to new tasks. We theoretically proved its convergence by reframing MAML and DMIL as hierarchical Bayes inference processes to get their equivalence. Empirically, we successfully acquire transferable hierarchical structures in both Meta-world and Kitchen Environments.

The limitations of DMIL comes from several aspects, and future works can seek meaningful extensions in these perspectives. Firstly, DMIL models all tasks as bi-level structures. However, in real-world situations, tasks may be multi-level structures. One can extend DMIL to multi-level hierarchical structures like done in recent works (Shu et al., 2018). Secondly, DMIL does not capture temporal information in demonstrations. Future state conditioning in Hakhamaneshi et al. (2021) seems an effective tool to improve few-shot imitation learning performance in long-horizon tasks such as in the Kitchen environments. Future works can employ temporal module such as transformer (Vaswani et al., 2017) as the high-level network of DMIL to improve its performance.

## 7 REPRODUCIBILITY STATEMENT

We provide video results in supplementary materials. We provide some basic codes to run few-shot testing in meta-world environments in an anonymous repository [1]. The complete training code will be provided later. Details of experiments such as environments, hyper-parameter setting, model setup, training procedure and data preprocessing can be found at E.

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

# A   ALGORITHM

---

**Algorithm 1** Dual Meta Imitation Learning

---

**Require:** task distribution $p(\mathcal{T})$, multi-task demonstrations $\{\mathcal{D}_i\}, i = 1, \cdots, m$, initial parameters of high-level network $\theta_h$ and sub-skill policies $\theta_{l1}, \cdots, \theta_{lK}$, inner and outer learning rate $\alpha, \beta$.

**while** not done **do**

    Sample batch of tasks $\mathcal{T}_i \sim p(\mathcal{T})$

    **for all** $\mathcal{T}_i$ **do**

        Sample $\{\tau_{i1}\}, \{\tau_{i2}\}, \{\tau_{i3}\}, \{\tau_{i4}\}$ from $\{\mathcal{D}_i\}$

        Evaluate $\nabla_{\theta_h}\mathcal{L}_h(\theta_h, \tau_{i1})$ according to 3 and $\tau_{i1}$

        Compute adapted parameters of high-level network: $\lambda_h = \theta_h - \alpha\nabla_{\theta_h}\mathcal{L}_h(\theta_h, \tau_{i1})$

        Evaluate $\nabla_{\theta_{lk}}\mathcal{L}_{BC}(\theta_{lk}, \mathcal{D}_{2k})$ according to 4 and $\tau_{i2}$, $k = 1, \cdots, K$

        Compute adapted parameters of sub-skills: $\lambda_{lk} = \theta_{lk} - \alpha\nabla_{\theta_{lk}}\mathcal{L}_{BC}(\theta_{lk}, \mathcal{D}_{2k})$, $k = 1, \cdots, K$

        Evaluate $\nabla_{\theta_h}\mathcal{L}_{\mathcal{T}_i}(\lambda_h, \tau_{i3})$ and $\nabla_{\theta_{lk}}\mathcal{L}_{\mathcal{T}_i}(\lambda_{lk}, \mathcal{D}_{4k})$, $k = 1, \cdots, K$

    **end for**

    Update $\theta_h \leftarrow \theta_h - \beta\nabla_{\theta_h}\sum_{\mathcal{T}_i \sim p(\mathcal{T})}\mathcal{L}_{\mathcal{T}_i}(\lambda_h, \tau_{i3})$

    Update $\theta_{lk} \leftarrow \theta_{lk} - \beta\nabla_{\theta_{lk}}\sum_{\mathcal{T}_i \sim p(\mathcal{T})}\mathcal{L}_{\mathcal{T}_i}(\lambda_{lk}, \mathcal{D}_{4k})$, $k = 1, \cdots, K$

**end while**

---

# B   AUXILIARY LOSS

We adopt an auxiliary loss for DMIL to better drive out meaningful sub-skills by punishing excessive switching of sub-skills along the trajectory. This comes from an intuitive idea: each sub-skill should be a temporal-extended macro-action, and the high-level policy only needs to switch to different skills few times along a task, as the same idea of *macro-action* in MLSH Frans et al. (2018). We denote:

$$sign(x) = \begin{cases} 1, & \text{if } x = True \\ 0, & \text{if } x = False \end{cases}, \tag{14}$$

and the auxiliary loss is:

$$\mathcal{L}_{aux}(\tau) = \sum_{t=0}^{T-1} sign(\hat{z}_{t+1} \neq \hat{z}_t) / len(\tau). \tag{15}$$

Although this operation seems discrete, in practice we can use the operations in modern deep learning framework such as PyTorch (Paszke et al., 2019) to make it differentiable. We add this loss function to the $\mathcal{L}_h(\theta_h, \tau_{i1})$ and $\nabla_{\theta_h}\mathcal{L}(\lambda_h, \tau_{i3})$ with a coefficient $\lambda = 1$. We also perform ablation studies of $\mathcal{L}_{aux}$ and results are in table 7.

# C   PROOFS

## C.1   PROOF OF LEMMA 1

**Lemma 1** In case $q(\phi_i; \lambda_i)$ is a Dirac-delta function and choosing Gaussian prior for $p(\phi_i|\theta)$, equation 8 equals to the inner-update step of MAML, that is, maximizing $\log p(\mathcal{D}_i^{tr})$ w.r.t. $\lambda_i$ by early-stopping gradient-ascent with choosing $\mu_\theta$ as initial point:

$$\lambda_i(\mathcal{D}_i^{tr}; \theta) = \mu_\theta + \alpha\nabla_\theta \log p\left(\mathcal{D}_i^{tr}|\theta\right)|_{\theta=\mu_\theta}. \tag{16}$$

*Proof:* in case of the conditions of Lemma 1, we have:

$$\lambda_i(\mathcal{D}_i^{tr}; \theta) = \arg\max_{\lambda_i}[\log p(\mathcal{D}_i^{tr}|\mu_{\lambda_i}) - \|\mu_{\lambda_i} - \mu_\theta\|^2 / 2\Sigma_\theta^2], \tag{17}$$

As stated in Grant et al. (2018), firstly in the case of linear models, early stopping of an iterative gradient descent process of $\lambda$ equals to the maximum posterior estimation (MAP) (Santos, 1996).

In our case the posterior distribution refers to $q(\phi_i|\lambda_i)$, and MAML is a Bayes process to find the MAP estimate as the point estimate of $\lambda_i(\mathcal{D}_i^{tr};\theta)$. In the nonlinear case, this point estimate is not necessarily the global mode of the posterior, and we can refer to Duvenaud et al. (2016) for another implicit posterior distribution over $\phi_i$ and making the early stopping procedure of MAML acting as priors to get the similar result.

## C.2 PROOF OF EQUATION 8

Equation 8 can be written as:

$$
\begin{aligned}
\lambda_i(\mathcal{D}_i^{tr}, \theta) &= \arg\min_{\lambda_i} \mathrm{KL}(q(\phi_i;\lambda_i)\|p(\phi_i|\mathcal{D}_i^{tr}, \theta)) \\
&= \arg\max_{\lambda_i} E_{q(\phi_i;\lambda_i)}[\log p(\phi_i|\mathcal{D}_i^{tr}, \theta) - \log q(\phi_i;\lambda_i)] \\
&= \arg\max_{\lambda_i} E_{q(\phi_i;\lambda_i)}[\log p(\mathcal{D}_i^{tr}|\phi_i)] - \mathrm{KL}(q(\phi_i;\lambda_i)\|p(\phi_i|\theta)),
\end{aligned}
\tag{18}
$$

where in MAML we assume $p(\mathcal{D}_i^{tr}|\phi_i) = p(\mathcal{D}_i^{tr}|\phi_i, \theta)$, and use the joint distribution $p(\mathcal{D}_i^{tr}, \phi_i|\theta)$ to replace $p(\phi_i|\mathcal{D}_i^{tr}, \theta)$ since we assume that $p(\mathcal{D}_i^{tr})$ subjects to uniform distribution. Thus 8 can be proved.

## C.3 PROOF OF THEOREM 1

**Theorem 1** In case that $\Sigma_\theta \to 0^+$, i.e., the uncertainty in the global latent variables $\theta$ is small, the following equation holds:

$$
\nabla_\theta \mathcal{L}(\theta, \lambda_1, \cdots, \lambda_M) = \sum_{i=1}^{M} \nabla_{\lambda_i} \log p(\mathcal{D}_i^{val}|\lambda_i) * \nabla_\theta \lambda_i(\mathcal{D}_i^{tr}, \theta).
\tag{19}
$$

*Proof:*

$$
\begin{aligned}
\nabla_\theta \mathcal{L}(\theta, \lambda_1, \cdots, \lambda_M) &\approx \sum_{i=1}^{M} \{\nabla_\theta E_{q(\phi_i;\lambda_i)}[\log p(\mathcal{D}_i, \phi_i|\theta) - \log q(\phi_i;\lambda_i)]\} \\
&= \sum_{i=1}^{M} \nabla_\theta [\log p(\mathcal{D}_i^{val}|\lambda_i(\mathcal{D}_i^{tr}, \theta)) - \log p(\lambda_i(\mathcal{D}_i^{tr}, \theta)|\theta)] \\
&\approx \sum_{i=1}^{M} \nabla_\theta \log p(\mathcal{D}_i^{val}|\lambda_i(\mathcal{D}_i^{tr}, \theta)) \\
&= \sum_{i=1}^{M} \nabla_{\lambda_i} \log p(\mathcal{D}_i^{val}|\lambda_i) * \nabla_\theta \lambda_i(\mathcal{D}_i^{tr}, \theta)
\end{aligned}
\tag{20}
$$

where the first approximate equal holds because the VI approximation error is small enough, and the second approximate equal holds because that in case $\Sigma_\theta \to 0^+$ and assuming $\lambda_i$ be a neuron network, $\log p(\lambda_i(\mathcal{D}_i^{tr}, \theta)|\theta) \approx 0$ holds almost everywhere, so $\nabla_\theta \log p(\lambda_i(\mathcal{D}_i^{tr}, \theta)|\theta) \approx 0$. Note the condition of theorem 1 is usually satisfied since we are using MAML, and the initial parameters $\theta$ are assumed to be deterministic.

From another perspective, the right side of above equation is the widely used meta-gradient in MAML, and it is equal to $\frac{1}{m} \sum_{i=1}^{m} \left(I - \alpha \nabla_\theta^2 \mathcal{L}_{BC}(\theta, \mathcal{D}_i^{tr})\right) * \nabla_{\lambda_i} \mathcal{L}_{BC}\left(\theta - \alpha \nabla_\theta \mathcal{L}_{BC}(\theta, \mathcal{D}_i^{tr}), \mathcal{D}_i^{val}\right)$, which is proved to be converged by Fallah et al. (2020).

## C.4 PROOF OF THEOREM 2

**Theorem 2** In case of $p(a_t|s_t, \theta_{lk}) \sim \mathcal{N}(\mu_{\theta_{lk}(s_t)}, \sigma^2)$, we have:

$$
\nabla_{\theta_h} \log p(\mathcal{D}_i^{tr}|\theta_h, \theta_{l1}, \cdots, \theta_{lK}) = \nabla_{\theta_h} \mathcal{L}_h(\theta_h, \mathcal{D}_i^{tr}),
\tag{21}
$$

and

$$\nabla_{\theta_{lk}} \log p(\mathcal{D}_i^{tr}|\theta_h, \theta_{l1}, \cdots, \theta_{lK}) = \nabla_{\theta_{lk}} \mathcal{L}_{BC}(\theta_{lk}, \mathcal{D}_{2k}), k = 1, \cdots, K. \tag{22}$$

*Proof:* since $p(\mathcal{D}_i^{tr}|\theta_h, \theta_{l1}, \cdots, \theta_{lK}) = \prod_{t=1}^N p(a_t|s_t, \theta_h, \theta_{l1}, \cdots, \theta_{lK})p(s_t|\theta_h, \theta_{l1}, \cdots, \theta_{lK})$ and the second term is independent of $\theta$, we consider the first conditional probability:

$$p(a_t|s_t, \theta_h, \theta_{l1}, \cdots, \theta_{lK}) = \sum_{k=1}^K p(z_k|s_t, \theta_h)p(a_t|s_t, \theta_{lk}). \tag{23}$$

In the HI step, $p(a_t|s_t, \theta_{lk})$ is fixed, thus 23 becomes a convex optimization problem:

$$\max_{\theta_h} \sum_{k=1}^K p(z_k|s_t, \theta_h)p(a_t|s_t, \theta) \tag{24}$$

The solution of this problem is $\lambda_h^*$ which satisfies $p(z_k|s_t, a_t, \lambda_h^*) = 1$, $k = \arg\max_k p(a_t|s_t, \theta_{lk})$. This means that $\pi_{\theta_h}$ needs to predict the sub-skill category at time step $t$ as $k$, in which case $\pi_{\theta_{lk}}$ can maximize $p(a_t|s_t, \theta_{lk})$. In case we choose $\pi_{\theta_h}$ to be a classifier that employs a Softmax layer at the end, minimizing the cross entropy loss 3 equals to maximize 24, thus 21 can be proved.

In the LI step, $p(z_k|s_t, \lambda_h)$ is fixed, and the data sets for optimizing $\theta_{l1}, \cdots, \theta_{lK}$ are also fixed as $\mathcal{D}_{2k} = \{(s_{ijt}, a_{ijt})|\hat{z}_{i2t} = k\}_{t=1}^{N_k}$. Thus we need to maximize each $p(a_t|s_t, \theta_{lk})$ with $\mathcal{D}_{2k}$. In case of $p(a_t|s_t, \theta_{lk}) \sim \mathcal{N}(\mu_{\theta_{lk}(s_t)}, \sigma^2) \propto \exp[-\frac{(a_t - \pi_{\theta_{lk}}(s_t))^2}{2\sigma^2}]$, we have $\max_{\theta_{lk}} p(a_t|s_t, \theta_{lk}) \Leftrightarrow \min_{\theta_{lk}}(a_t - \pi_{\theta_{lk}}(s_t))^2$, which leads to the loss function 4, thus 22 can be proved, which finishes the prove of Theorem 2.

## C.5  PROOF OF THE E-STEP OF DMIL

According to Theorem 1, we aim to maximize 17 w.r.t $\lambda_i$ from the initial point $\theta_i$ with coordinate gradient ascent. We here need to prove that in DMIL, we could also achieve the global maximum point of $\lambda_i$ as in MAML. We first give out the following Lemma:

**Lemma 2** Let $x$ be the solution found by coordinate gradient descent of $f(x)$. Let $x_i, i = 1, \cdots, n$ be the $n$ coordinate directions used in the optimization process. If $f(x)$ can be decomposed as:

$$f(x) = g(x) + \sum_{i=1}^n h_i(x), \tag{25}$$

where $g(x)$ is a differentiable convex function, and each $h_i(x)$ is a convex function of the coordinate direction $x_i$, then $x$ is the global minimum of $f(x)$.

*Proof:* Let $y$ be another arbitrary point, we have:

$$\begin{aligned} f(y) - f(x) &= g(y) + h(y) - (g(x) + h(x)) \\ &\geq \nabla_x g(x)^T(y - x) + \sum_{i=1}^n h_i(y_i) - h_i(x_i) \\ &= \sum_{i=1}^n (\nabla_i g(x)(y_i - x_i) + h_i(y_i) - h_i(x_i)) \\ &\geq 0. \end{aligned} \tag{26}$$

Now let's consider our problem. Consider

$$
\begin{aligned}
\log p(\mathcal{D}_i^{tr}|\theta_{ih}, \theta_{il}) &= \log \sum_{t=1}^{T} p(a_t|s_t, \theta_{ih}, \theta_{il})p(s_t) \\
&= \log \sum_{t=1}^{T} p(s_t) \sum_{k=1}^{K} p(z_k|s_t, a_t, \theta_{ih})p(a_t|s_t, \theta_{il}) \\
&\geq \sum_{t=1}^{T} [\log p(s_t) + \log \sum_{k=1}^{K} p(z_k|s_t, a_t, \theta_{ih})p(a_t|s_t, \theta_{il})] \qquad (27) \\
&\geq \sum_{t=1}^{T} [\log p(s_t) + \sum_{k=1}^{K} \log p(z_k|s_t, a_t, \theta_{ih})p(a_t|s_t, \theta_{il})] \\
&= \sum_{t=1}^{T} [\log p(s_t) + \sum_{k=1}^{K} \log p(z_k|s_t, a_t, \theta_{ih}) + \sum_{k=1}^{K} \log p(a_t|s_t, \theta_{il})].
\end{aligned}
$$

In our case, two coordinate directions are $\theta_{ih}$ and $\theta_{il}$. Let's consider the terms inside the brackets. According to Lemma 2, we can think $\log p(s_t)$ as $g(x)$ (here it equals to constant), $\sum_{k=1}^{K} \log p(z_k|s_t, a_t, \theta_{ih})$ as $h_1(x)$ and $\sum_{k=1}^{K} \log p(a_t|s_t, \theta_{il})$ as $h_2(x)$. Thus the optimum can be proved.

## D    ADDITIONAL ABLATION STUDIES

### D.1    EFFECTS OF THE BI-LEVEL META-LEARNING PROCESS

We use two variants of DMIL to see the effectiveness of bi-level meta-earning process. **DMIL-High**: a variant that only meta-learns the high-level network, and **DMIL-Low**: a variant that only meta-learns sub-skills. We use Option-GAIL as a comparison that does not meta-learn any level of the hierarchical structure.

Table 4 shows the results of this ablation study. DMIL-High achieves close results with OptionGAIL in all meta-training suites and better results in all meta-testing suites, but worse than DMIL in all cases. This shows that meta-learning the high-level network can help the hierarchical structure adapt to new tasks, but only transferring the high-level network is not enough for accomplish all kinds of new tasks. DMIL-Low achieves poor results in all suites except in ML10 meta-training suites. This shows that transferring the high-level network is necessary when training on a large scale of tasks or testing in new tasks.

Table 4: Success rates of DMIL-High, DMIL-Low, DMIL and OptionGAIL on Meta-world environments with $K = 3$. Each data point comes from the success rate of 20 tests.

| | ML10 | | | | ML45 | | | |
|---|---|---|---|---|---|---|---|---|
| | Meta-training | | Meta-testing | | Meta-training | | Meta-testing | |
| Methods | 1-shot | 3-shot | 1-shot | 3-shot | 1-shot | 3-shot | 1-shot | 3-shot |
| OptionGAIL | 0.755±0.011 | **0.952±0.016** | 0.241±0.042 | 0.640±0.025 | 0.506±0.008 | 0.715±0.006 | 0.220±0.013 | 0.481±0.010 |
| DMIL-High | 0.634±0.001 | 0.914±0.011 | 0.298±0.012 | 0.670±0.015 | 0.495±0.007 | 0.735±0.006 | 0.280±0.016 | 0.551±0.011 |
| DMIL-Low | 0.746±0.011 | 0.943±0.006 | 0.291±0.024 | 0.666±0.021 | 0.511±0.005 | 0.765±0.009 | 0.266±0.010 | 0.492±0.014 |
| DMIL | **0.775±0.010** | 0.949±0.009 | **0.396±0.016** | **0.710±0.021** | **0.590±0.010** | **0.859±0.008** | **0.376±0.004** | **0.640±0.009** |

For better understand the effect of bi-level meta-learning process, we perform ablation study for three DMIL variants in an manually-designed new task *push-around-wall* (figure 8). In this task, the robot needs to grasp a cube and circle it around the wall. This is a brand new skill that is not in the meta-world suite. The accomplishment of this new task requires quickly adapting abilities of both the high-level network and sub-skills. We sample two demonstrations and use the first one as few-shot data, and illustrate the sub-skill categories of the second demonstration given by the high-level network at figure 4(b). Before adaptation, all variants give out approximately random results. However, after one-shot adaptation, DMIL classifies almost every state into sub-skill 0 and

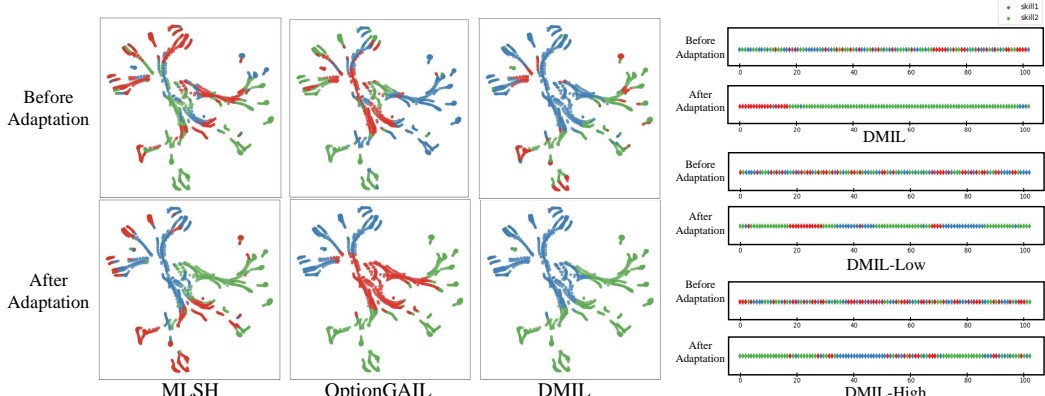

(a) T-sne results of demonstrations and sub-skill categories of several hierarchical models for meta-testing task *hand-insert*.

(b) Sub-skill categories of task *push-around-wall* for ablation study.

Figure 4: T-sne results and ablation studies about the bi-level meta-learning process.

sub-skill 2, which indicates these two sub-skills in DMIL have been adapted to the new task, and the high-level network has also been adapted with the supervision from adapted sub-skills. Compared to DMIL, DMIL-Low still can not give out reasonable results after adaptation, since its high-level network lacks the ability to quickly transfer to new tasks. Instead, DMIL-High gives out plausible results after adaptation. This shows the high-level network has adapted to the new task according to the supervision from adapted sub-skills, but no sub-skill can dominate for a long time period since all sub-skills lack the quickly adaptation ability.

### D.2 T-SNE RESULTS OF DIFFERENT METHODS

For comparison of different methods, we illustrate the t-sne results of states of each sub-skill in an ML45 meta-testing task hand-insert in figure 4(a). We use 3 demonstrations for adaptation, and draw the t-sne results on another 16 demonstrations. This task is a meta-testing task, so no method has ever been trained on this task before.

MLSH shows almost random clustering results no matter before and after adaptation, since its high-level network is relearned in new tasks. OptionGAIL clusters to three sub-skills after adaptation. Compared to them, DMIL clusters the data to only two sub-skills after adaptation. We believe fewer categories reflect more meaningful sub-skills are developed in DMIL.

### D.3 EFFECTS OF SUB-SKILL NUMBER K IN THE KITCHEN ENVIRONMENT

We perform ablation studies of sub-skill number $K$ on the Kitchen environments and choose $K = 2, 4, 8$ respectively. Table 5 shows the results. We can see that a smaller number of sub-skills can achieve better results on such four unseen results that a large number of sub-skills. This may indicate that the sub-skill number $K$ can work as a 'bottleneck' like the middle layer in an auto-encoder.

Table 5: Ablations of sub-skill number $K$ in Kitchen environments.

| Task (Unseen) | K=2 | K=4 | K=8 |
|---|---|---|---|
| Microwave, Kettle, Top Burner, Light Switch | **1.9±0.43** | 1.5±0.48 | 1.7±0.22 |
| Microwave, Bottom Burner, Light Switch, Slide Cabinet | 2.15±0.19 | **2.35±0.39** | 2.0±0.37 |
| Microwave, Kettle, Hinge Cabinet, Slide Cabinet | 2.45±0.25 | **3.15±0.22** | 1.85±0.23 |
| Microwave, Kettle, Hinge Cabinet, Slide Cabinet | 2.01±0.24 | **2.95±0.44** | 2.44±0.47 |

## D.4 EFFECTS OF FINE-TUNING STEPS

As all few-shot learning problems, the fine-tuning steps in new tasks to some extent determine the performance of the trained model. It controls the balance between under-fitting and over-fitting. We perform ablation studies of fine-tune steps in Meta-world benchmarks with $K = 5$ and lr=1e-2 in table 6. Results are as follows:

Table 6: Ablation studies of the fine-tuning steps in Meat-world experiments with $K = 5$.

| | ML10 | | | | ML45 | | | |
| | Meta-training | | Meta-testing | | Meta-training | | Meta-testing | |
| fine-tune steps | 1-shot | 3-shot | 1-shot | 3-shot | 1-shot | 3-shot | 1-shot | 3-shot |
|---|---|---|---|---|---|---|---|---|
| 10 | 0.5 | 0.575 | **0.28** | 0.24 | 0.374 | 0.49 | 0.05 | 0.17 |
| 30 | 0.69 | **0.915** | 0.25 | 0.31 | 0.602 | 0.85 | 0.13 | 0.32 |
| 50 | **0.695** | 0.895 | 0.27 | 0.39 | 0.583 | 0.87 | 0.12 | 0.32 |
| 100 | 0.665 | 0.905 | 0.23 | 0.41 | 0.614 | 0.872 | 0.07 | 0.43 |
| 300 | 0.66 | 0.845 | **0.28** | **0.44** | **0.605** | **0.876** | 0.12 | **0.44** |
| 500 | 0.63 | 0.91 | 0.25 | 0.39 | 0.584 | 0.867 | **0.13** | 0.42 |
| Range | 0.195 | 0.34 | 0.05 | 0.2 | 0.231 | 0.386 | 0.08 | 0.27 |

## D.5 EFFECTS OF CONTINUITY REGULARIZATION

We perform ablation studies of the effect of continuity regularization as following table 7. DMIL_nc means no continuity regularization. We can see that the continuity constraint would damage meta-training performance slightly, but increase the meta-testing performance greatly.

Table 7: $K = 10$

| | ML10 | | | | ML45 | | | |
| | Meta-training | | Meta-testing | | Meta-training | | Meta-testing | |
| variants | 1-shot | 3-shot | 1-shot | 3-shot | 1-shot | 3-shot | 1-shot | 3-shot |
|---|---|---|---|---|---|---|---|---|
| DMIL | **0.795** | 0.94 | **0.52** | **0.57** | **0.713** | 0.92 | **0.21** | **0.48** |
| DMIL_nc | 0.788 | **0.96** | 0.32 | 0.56 | 0.703 | **0.927** | 0.17 | 0.35 |
| Gap | 0.007 | -0.02 | 0.2 | 0.01 | 0.01 | -0.007 | 0.04 | 0.13 |

## D.6 EFFECTS OF HARD/SOFT EM CHOICES

In DMIL, we use hard EM algorithm to train the high-level network. One may think about to use soft cross entropy loss to train the high-level network to get better results. We perform this ablation study in the following table 8. We can see that a soft cross entropy training won't help increase the whole success rates. This may comes from that, usually we use soft cross entropy (such as label smoothing) to prevent over-fitting. However, in our situation, this may cause under-fitting, since training on such a large scale of diverse manipulation tasks is already very difficult. Future works can seek more comparisons about this choice.

Additionally, we found an interesting phenomena that the training loss of the high-level network with a softmax shows a trend of rising first and then falling, as shown in . In our experiments, a softmax loss may regularize the optimization process to make the high-level network be under-fitting on training data. This may come from that, the experiment environment (Meta-world) contains a large scale of manipulation tasks, in which the training of the high-level network can be difficult and unstable. Thus a soft-max cross entropy loss cannot help that much here like how it works as a regularizer to prevent over-fitting in the label smoothing (Müller et al., 2019).

Table 8: Ablation about hard/soft EM choices with $K = 5$ in the Meta-world environments.

| | ML10 | | | | ML45 | | | |
| | Meta-training | | Meta-testing | | Meta-training | | Meta-testing | |
| variants | 1-shot | 3-shot | 1-shot | 3-shot | 1-shot | 3-shot | 1-shot | 3-shot |
|---|---|---|---|---|---|---|---|---|
| DMIL | **0.795** | **0.94** | **0.52** | **0.57** | **0.713333** | **0.92** | **0.21** | **0.48** |
| DMIL_soft | 0.37 | 0.65 | 0.33 | 0.46 | 0.235556 | 0.43 | 0.1 | 0.32 |
| Gap | 0.425 | 0.29 | 0.19 | 0.11 | 0.477778 | 0.49 | 0.11 | 0.16 |

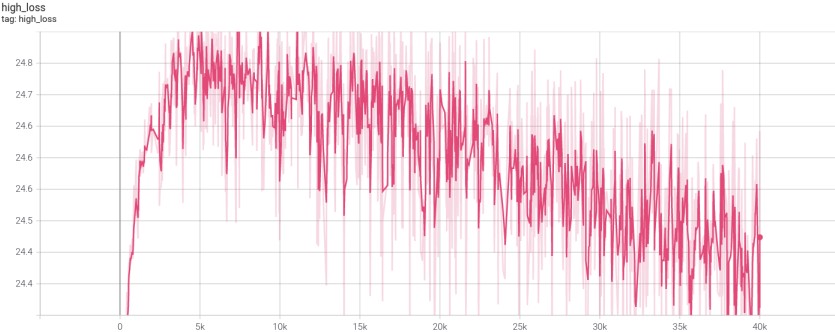

Figure 5: The training loss of the high-level network with a softmax shows a trend of rising first and then falling.

# E EXPERIMENT DETAILS

## E.1 ENVIRONMENTS

See figure 7,6,9 and 8.

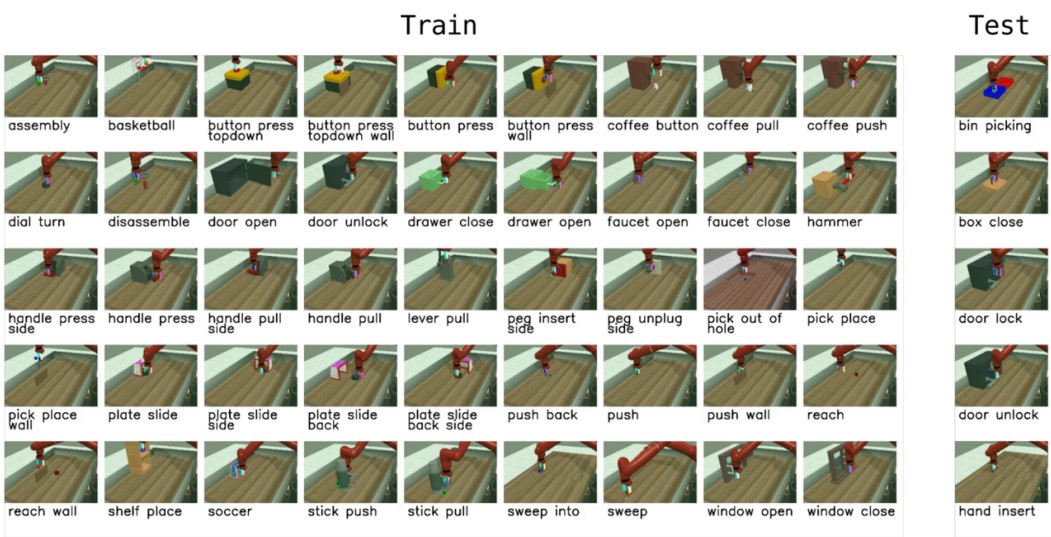

Figure 6: The ML45 environment.

## E.2 MODEL SETUP

**DMIL:** The high-level network and each sub-skill is modeled with a 4-layer fully-connected neuron network, with 512 ReLU units in each layer. We use Adam as the meta-optimizer. **DMIL-High** and

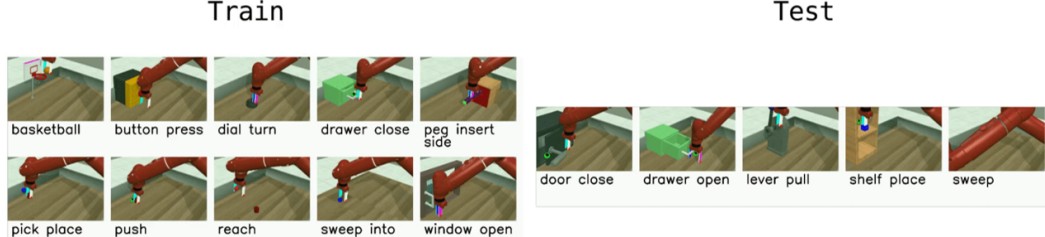

Figure 7: The ML10 environment.

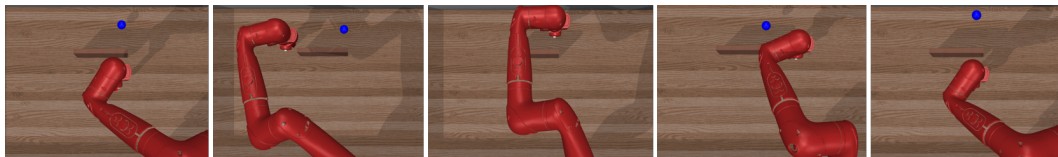

Figure 8: Task *push-around-wall*.

Table 9: DMIL hyper-parameters.

| Parameter | Value |
|---|---|
| $\alpha$ | 5e-4 |
| $\beta$ | 1e-4 |
| fine-tune iterations | 3 |
| batch size (in trajectory) | 16 |
| $\lambda$ | 0.1 |

Table 10: MIL hyper-parameters.

| Parameter | Value |
|---|---|
| $n_{head}$ | 8 |
| $n_{layer}$ | 3 |
| $d_{model}$ | 512 |
| $d_k$ | 64 |
| $d_v$ | 64 |
| $n_{position}$ | 250 |
| dropout | 0.1 |
| batch size (in state-action pair) | 512 |

**DMIL-Low** use the same architecture with DMIL. Hyper-parameter settings are available in table 9.

**MIL:** We use a transformer (Vaswani et al., 2017) as the policy to perform MAML. The input of the encoder is the whole one-shot demonstration or 3-shot demonstrations with concatenated state and action. The input of the decoder is current state. The output is the predicted action. Hyper-parameter settings are available in table 10.

**MLSH:** We use the same settings of network with DMIL here. The macro step of high-level network is 3. Since our problem is not a reinforcement learning process, we use the *behavior cloning* variant of MLSH. The pseudo reward is defined by the negative mean square loss of the predicted action and the ground truth, and we perform pseudo reinforcement learning process with off-policy demonstration data. We use PPO as our reinforcement learning algorithm. Note in this way we ignore the importance sampling weights that required by replacing the sampling process in the environments with the demonstrations in the replay buffer, which has been shown to be effective in practice in

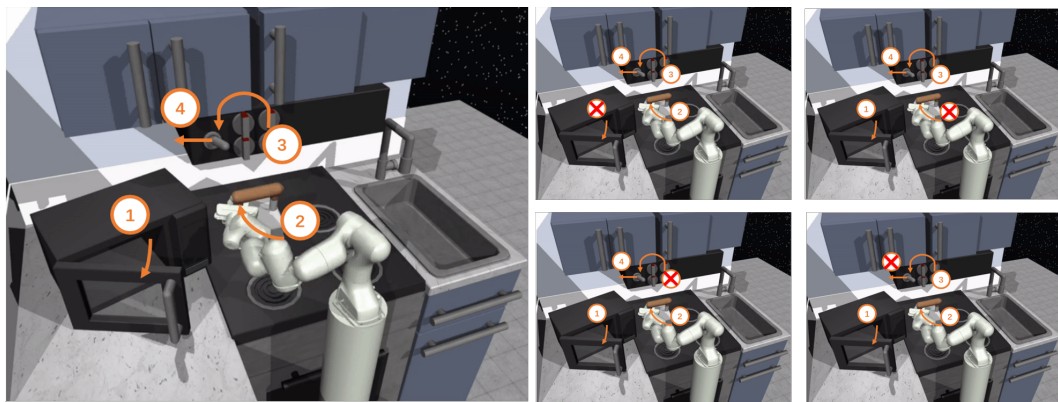

Figure 9: Kitchen environments.

Table 11: MLSH hyper-parameters.

| Parameter | Value |
|---|---|
| high-level learning rate | 1e-3 |
| sub-skill learning rate | 1e-4 |
| PPO clip threshold | 0.02 |
| high-level warmup step | 500 |
| joint update step | 1000 |
| batch size (in state-action pair) | 900 |

Table 12: PEMIRL hyper-parameters.

| Parameter | Value |
|---|---|
| learning rate of all models | 1e-4 |
| PPO clip threshold | 0.02 |
| coefficient ($\gamma$) of $h_\phi$ | 1 |
| $\beta$ | 0.1 |
| batch size (in trajectory) | 16 |

Kostrikov et al. (2018); Ghasemipour et al. (2019). Hyper-parameter settings are available in table 11.

**PEMIRL:** We use the same setting of the high-level network as the policy $\pi_\omega$ and the inference model $q_\psi$ in PEMIRL. We use PPO as our reinforcement learning algorithm. We use a 3-layer fully-connected neuron network as the context-dependent disentangled reward estimator $r_\theta(s, m)$ and the context-dependent potential function $h_\phi(s, m)$. Here we also use the *behavior cloning* variant of PEMIRL to only train models on the off-policy data. Hyper-parameter settings are available in table 12.

### E.3 TRAINING DETAILS

For fine-tuning, OptionGAIL, DMIL-Low and DMIL-High have no meta-learning mechanism for (some parts of) the trained model. In the few-shot adaptation process, we have different fine-tune method for these baselines:

**OptionGAIL:** We train OptionGAIL models on the provided few-shot demonstrations for a few epochs.

**DMIL-Low:** We fix the high-level network and only fine-tune sub-skills with few-shot demonstrations.

**DMIL-High**: We fix sub-skills and only fine-tune the high-level network with few-shot demonstrations.

### E.4    DETAILED RESULTS

We provided detailed success rates of each environments of different methods in figure 10 and 11 with $K = 3$.

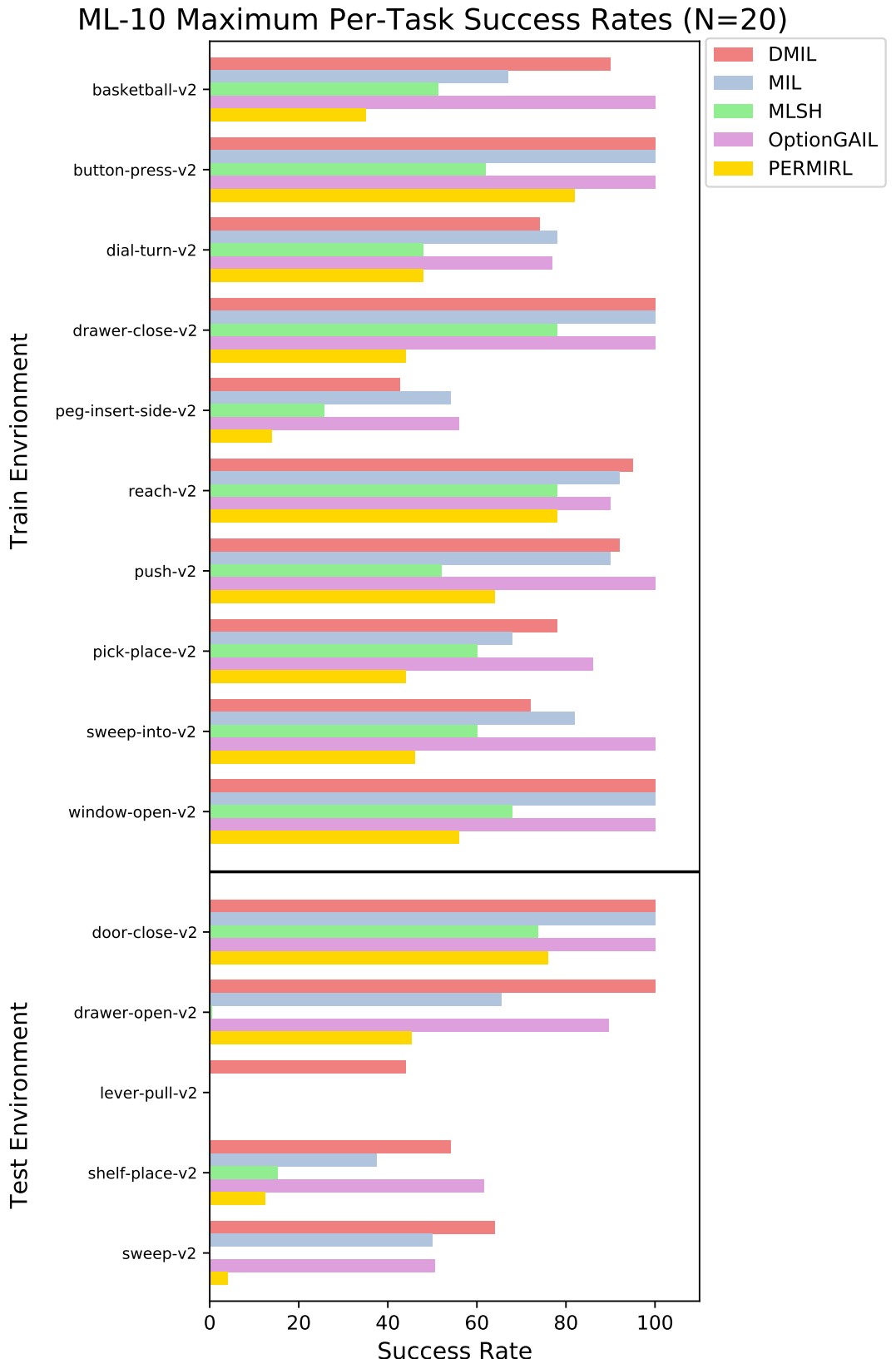

Figure 10: The ML10 results of different methods after 3-shot adaptation.

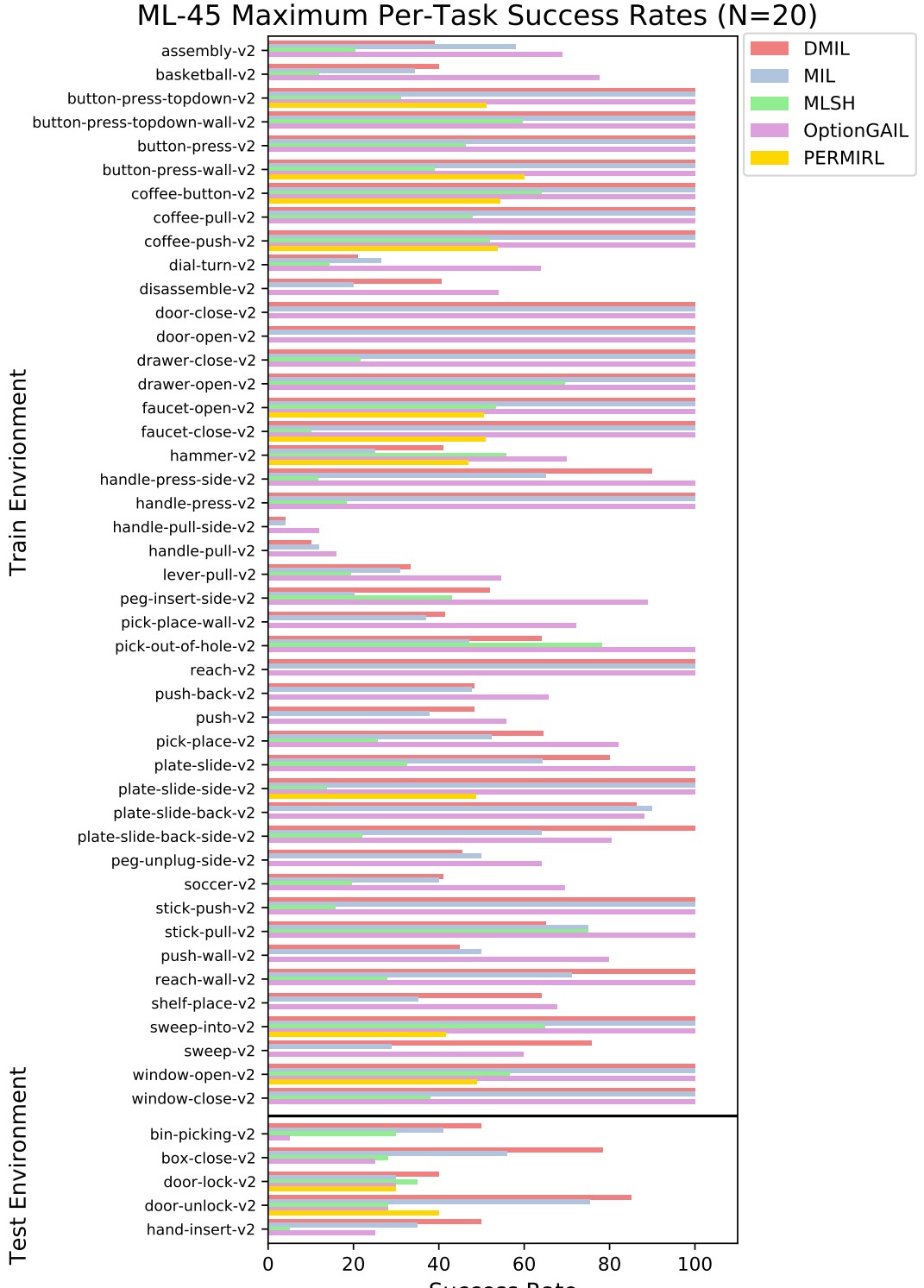

Figure 11: The ML45 results of different methods after 3-shot adaptation.

