# OpenReview forum: "Transferring Hierarchical Structure with Dual Meta Imitation Learning"
_ICLR.cc/2022/Conference — ICLR 2022 Submitted_

### Official Review · Reviewer_wfsx · 2021-10-30

**Correctness:** 3
**Technical Novelty And Significance:** 3
**Empirical Novelty And Significance:** 3
**Recommendation:** 6
**Confidence:** 3

**Main Review:**

## Strengths
- few-shot imitation is an interesting and impactful problem

- the paper solves a hard problem: jointly learning HL and LL policy + quickly adapting them on unseen tasks

- the method is clearly described, the figure help understanding the approach

- the method is evaluated on a representative benchmark and compared to meaningful baselines

- the qualitative results show that the approach learns meaningful skills in the MetaWorld benchmark


## Weaknesses

- My main concern with the method is that it is trying to solve too many things at once: (1) inferring skills from unstructured datasets, (2) jointly learning high-level and low-level policy and (3) meta-learning both these policies to have them adapt quickly on new tasks. Intuitively, the resulting iterated meta-optimization scheme seems unstable to train / sensitive to choices like the policy initialization, number of subskills, switching regularization coefficient, ... (the theoretical convergence results do not elucidate how hard it is to train agents with this approach *in practice*). The paper could add an analysis on hyperparameter robustness of training to show the stability of the approach.

- While the paper cites many works on hierarchical imitation and meta-imitation learning, it does not cite the relevant recent works on learning skills from large, multi-task datasets (SPiRL, Pertsch et al 2020, OPAL, Ajay et al 2021) and using them for imitation learning (SkiLD, Pertsch et al 2021, FIST, Hakhamaneshi et al 2021). Especially FIST seems like an interesting method to compare to since it is made for the few-shot imitation setting. That being said: this comparison is not mandatory since FIST is concurrent work.

- I think the paper could be strengthened by including a comparison to a method that is hierarchical but *does not* learn HL and LL policies jointly. All of the abovementioned works first learn a set of low-level skills and then learn a high-level policy over them. It would improve the comparisons of this paper to include an approach that pre-trains temporally extended low-level skills, then freezes them and meta-trains a high-level policy over these skills. Such a training scheme would presumably be much easier to tune (since it does not require EM-style iterated training) -- so it would be good to see whether the performance is much worse than the proposed, more complicated method. (the already included DIML-High ablation still trains HL and LL jointly even though it keeps the LL fixed during downstream finetuning from what I understand)

- The paper does not include an ablation on the chosen number of discrete learned skills. It is possible that the DIML-High ablation could perform better if it had a larger number of low-level skills to choose from (since it does not adapt the low-level skills to the downstream task), so including such an experiment would strengthen the paper.


## Further Suggestions
- I wonder whether the approach could apply an idea similar to label smoothing where instead of using a hard max for the inner training loop of the HL policy we would set the targets for the HL cross entropy loss to be the softmax of the action prediction errors of the low-level. Maybe such a more nuanced training signal could improve the training stability?

**Summary Of The Paper:**

The paper proposes an approach for few-shot imitation learning that jointly meta-learns a high-level policy and a set of low-level policies from a diverse set of demonstrations (multi-task). It then finetunes both high-level and low-level policies on few demonstrations of the target task for imitating them. In experiments on the MetaWorld50 task suite the proposed approach outperforms few-shot imitation methods that either don't use hierarchy or don't use meta-learning.


**Summary Of The Review:**

The paper proposes an interesting approach for jointly learning adaptable high and low-level skills. While the method is well-explained it seems to more complex than alternative approaches and the current experiments don't fully convince me that it would be easy to train on new tasks without a significant amount of tuning to get the iterated learning loop to converge. If the authors can provide experimental evidence that the method is no harder to train / tune than alternative approaches for few-shot imitation I am leaning towards accepting the submission, but I am open to changing my opinion based on the other reviewer's feedback.

---

> ### Author Response · Authors · 2021-11-21
> **Response to Reviewer wfsx (2/2)**
>
> **Q3: I think the paper could be strengthened by including a comparison to a method that is hierarchical but does not learn HL and LL policies jointly.**
>
> A3: Thanks for your suggestions. Actually, we think this is a very interesting question. Separately learning the HL and LL policies needs to first discover a set of sub-skills and then learn the high-level network. The problem is that any unsupervised skill discovery method (a general clustering method) can be employed to get skills from unsegmented demonstrations, such as EM-like algorithms in DMIL, information-theoretic methods (like [5]) and metric-learning methods (like [4]), and the pros and cons between them have not been studied yet. The motivation of this paper is to combine one of them with meta-learning to get transferable sub-skills that can adapt to new tasks. Among these methods, only EM-like methods directly learn a hierarchical structure from unsegmented data, which is more elegant. Thus we choose it in our paper and tackle several challenges of incorporating it with MAML.
>
> For your question, you can see the MLSH baseline as a kind of method that learns HL and LL policies separately. In MLSH, sub-skills are first driven by a hierarchical structure, then the HL policy is abandoned in every new task and relearned separately for a warm-up period. The comparison results are in Table 1 of our paper.
>
> **Q4: The paper does not include an ablation on the chosen number of discrete learned skills.**
>
> A4: We answer this question in A1 and A2.
>
> **Q5: Instead of using a hard max for the inner training loop of the HL policy, it might be better to use softmax of the action prediction errors of the low-level to make the targets for the HL cross entropy loss.**
>
> A5: Thanks for your suggestions. We add additional experiments to compare the hard max and soft max settings in Meta-world benchmarks. The results are as follows. From these results, we can see that a softmax optimization method cannot lead to better results.
>
> |  | ML10 | ML10 | ML10 | ML10 | ML45 | ML45 | ML45 | ML45 |
> |---|---|---|---|---|---|---|---|---|
> |  | Meta-training |  | Meta-testing |  | Meta-training |  | Meta-testing |  |
> | K | 1-shot | 3-shot | 1-shot | 3-shot | 1-shot | 3-shot | 1-shot | 3-shot |
> | 2 | **0.76** | **0.955** | **0.32** | **0.72** | **0.563** | **0.818** | **0.44** | **0.67** |
> | 3 | 0.775 | 0.949 | 0.396 | 0.71 | 0.59 | 0.859 | 0.376 | 0.64 |
> | 5 | 0.795 | 0.94 | 0.52 | 0.57 | 0.713 | 0.92 | 0.21 | 0.48 |
> | 10 | 0.8 | 0.975 | 0.38 | 0.62 | 0.736 | 0.931 | 0.34 | 0.64 |
>
> Additionally, we found an interesting phenomena that the training loss of the high-level network with a softmax loss shows a trend of rising first and then falling as shown in figure 5 in appendix D. In our experiments, a softmax loss may regularize the optimization process to make the high-level network be under-fitting on training data. This may come from that, the experiment environment (Meta-world) contains a large scale of manipulation tasks, in which the training of the high-level network can be difficult and unstable. Thus a soft-max cross entropy loss cannot help that much here  like how it works as a regularizer to prevent over-fitting in the label smoothing [6].
>
> ----------
>
> [1] Pertsch K, Lee Y, Lim J J. Accelerating reinforcement learning with learned skill priors[J]. arXiv preprint arXiv:2010.11944, 2020.
>
> [2] Ajay A, Kumar A, Agrawal P, et al. Opal: Offline primitive discovery for accelerating offline reinforcement learning[J]. arXiv preprint arXiv:2010.13611, 2020.
>
> [3] Karl Pertsch, Youngwoon Lee, Yue Wu and Joseph J. Lim. Demonstration-Guided Reinforcement Learning with Learned Skills[J]. arXiv preprint arXiv:2107.10253, 2021.
>
> [4] Hakhamaneshi K, Zhao R, Zhan A, et al. Hierarchical few-shot imitation with skill transition models[J]. arXiv preprint arXiv:2107.08981, 2021.
>
> [5] Eysenbach B, Gupta A, Ibarz J, et al. Diversity is all you need: Learning skills without a reward function[J]. arXiv preprint arXiv:1802.06070, 2018.
>
> [6] Müller R, Kornblith S, Hinton G. When does label smoothing help?[J]. arXiv preprint arXiv:1906.02629, 2019.

---

> > ### Comment · Reviewer_wfsx · 2021-11-28
> > **Rebuttal Response**
> >
> > Thank you for the efforts that went into this comprehensive rebuttal. I appreciate the added experiments in the kitchen environment and the comparisons to SPiRL and FIST. I will maintain my vote and suggest acceptance of the submission.
> >
> > Regarding the suggested experiment that does not jointly train high-level and low-level: in my understanding MLSH does not leverage offline multi-task data for learning skills but instead learns them via online interactions? Since the proposed approach learns skills from the offline expert data, I think a comparison to a baseline that also learns the skills from offline data, but *not* jointly with the high-level policy would be interesting.

---

> > > ### Author Response · Authors · 2021-11-30
> > > **Response to the Response of Reviewer wfsx**
> > >
> > > Thanks for your appreciation of our work. For the MLSH problem, in our paper, we are in an imitation learning scenario, so we use an offline variant of MLSH, which is introduced in detail in Appendix E.2 (at the bottom of page 21). We are sorry that we did not make this clear in the main text of this version. We will revise the experiment section in the future version.

---

> ### Author Response · Authors · 2021-11-21
> **Response to Reviewer wfsx (1/2)**
>
> Dear reviewer wfsx,
>
> We sincerely appreciate your valuable and insightful comments. We found them extremely helpful for improving our manuscript. We address each comment in detail, one by one below. We recommend you to check our new PDF for more details.
>
> **Q1: The resulting iterated meta-optimization scheme seems unstable to train / sensitive to choices like the policy initialization, number of sub-skills, switching regularization coefficient. The paper could add an analysis on hyper-parameter robustness of training to show the stability of the approach.**
>
> A1: Thanks for your suggestions. We perform additional experiments to give out more detailed analysis and experiment results on hyper-parameter setting, including the number of skills $ K $, the length of fine-tuning process, and the effect of continuous regularization in ablation studies section. Please check the ablation studies section of our new PDF for these results.
>
> **Q2: While the paper cites many works on hierarchical imitation and meta-imitation learning, it does not cite the relevant recent works on learning skills from large, multi-task datasets ([1] and [2]) and using them for imitation learning ([3] and [4]). Especially FIST seems like an interesting method to compare to since it is made for the few-shot imitation setting.**
>
> A2: Thanks for your suggestions. We perform additional experiments to compare DMIL with [4] in the Kitchen environments. Results are as follows:
>
> 1. We perform DMIL on the Kitchen environment as the way in [2], and the table below shows the rewards of different methods on four unseen tasks in the Kitchen environment. FIST-no-FT refers to a variant of FIST [2] that does not use future conditioning, which makes the comparison more fair. DMIL achieves higher rewards on two out of four tasks and comparable results on the other two tasks, which exhibits the effectiveness of the bi-level meta-training procedure. The poor performance of DMIL on the first task may comes from the choice of skill number $ K $ or from low-quality demonstrations. We perform ablation studies on $ K $ in the next table.
>
> | Task (Unseen) |  FIST-no-FT | SPiRL       | DMIL(ours) |
> |---|---|---|---|
> | Microwave, Kettle, **Top Burner**, Light Switch           | 2.0 ± 0.0   | **2.1 ± 0.48**  | 1.5±0.48   |
> | **Microwave**, Bottom Burner, Light Switch, Slide Cabinet | 0.0 ± 0.0   |  2.3 ± 0.49 | **2.35±0.39**  |
> | Microwave, **Kettle**, Hinge Cabinet, Slide Cabinet       | 1.0 ± 0.0   | 1.9 ± 0.29  | **3.15±0.22**  |
> | Microwave, Kettle, Hinge Cabinet, **Slide Cabinet**       | 2.0 ± 0.0   | **3.3 ± 0.38**  | 2.95±0.44  |
>
> 2. We perform ablation studies on skill number $ K $ in both Meta-world benchmarks and the kitchen environments, and results are as follows. In the Meta-world environments, we can see that a larger $ K $ can lead to higher success rates on meta-training tasks, but a smaller $ K $ can lead to better results on meta-testing tasks. This tells us that an excessive number of sub-skills can result in over-fitting of training data, and a smaller $ K $ can play the role of regularization. In the Kitchen environments, a larger number of skills can not lead to better testing results in unseen tasks.
>
> |  | ML10 | ML10 |ML10  |ML10  | ML45 |ML45  | ML45 | ML45 |
> |---|---|---|---|---|---|---|---|---|
> |  | Meta-training | Meta-training | Meta-testing | Meta-testing | Meta-training | Meta-training | Meta-testing | Meta-testing |
> | K | 1-shot | 3-shot | 1-shot | 3-shot | 1-shot | 3-shot | 1-shot | 3-shot |
> | 2 | 0.76 | 0.955 | 0.32 | **0.72** | 0.563 | 0.818 | **0.44** | **0.67** |
> | 3 | 0.775 | 0.949 | 0.396 | 0.71 | 0.59 | 0.859 | 0.376 | 0.64 |
> | 5 | 0.795 | 0.94 | **0.52** | 0.57 | 0.713 | 0.92 | 0.21 | 0.48 |
> | 10 | **0.8** | **0.975** | 0.38 | 0.62 | **0.736** | **0.931** | 0.34 | 0.64 |
>
>
>
> | Task (Unseen) |  K=2 | K=4 | K=8 |
> |---|---|---|---|
> | Microwave, Kettle, **Top Burner**, Light Switch | **1.9±0.43** | 1.5±0.48 | 1.7±0.22 |
> | **Microwave**, Bottom Burner, Light Switch, Slide Cabinet | 2.15±0.19 | **2.35±0.39** | 2.0±0.37 |
> | Microwave, **Kettle**, Hinge Cabinet, Slide Cabinet | 2.45±0.25 | **3.15±0.22** | 1.85±0.23 |
> | Microwave, Kettle, Hinge Cabinet, **Slide Cabinet** | 2.01±0.24 | **2.95±0.44** | 2.44±0.47 |

---

### Official Review · Reviewer_ouLD · 2021-10-31

**Correctness:** 4
**Technical Novelty And Significance:** 3
**Empirical Novelty And Significance:** 2
**Recommendation:** 6
**Confidence:** 4

**Main Review:**

This paper has a number of strengths:
1. This paper studies an important research problem  — the ability to discover meaningful abstractions of behavior from prior experience and to transfer these abstractions to efficiently learn new tasks
2. The related work section lays out a detailed overview of relevant work
3. While the proposed approach integrates existing elements from MAML and hierarchical imitation learning, it does so in a unique and novel manner
4. The theoretical analysis complements the work well (the AC should note however — I did not scrutinize this section sufficiently to verify its claims)
5. The experiments consider a thorough set of baselines

Alongside these strengths, I also have numerous concerns, all regarding the quality and clarity of presentation:
1. In my view, the introduction of the paper sells the work short. Much of the discussion is on centered “what” approaches previously exist and “what” the technical components of this paper are. Relatively little discussion in centered on “why” this proposed research problem is important, “why” previous approaches falls short, and “why” addressing the limitations of prior approaches is challenging. Adding this context can help better motivate this work.
2. The figures are difficult to follow. In particular, figure 1 and 2 have a lot of visual details, but are accompanied with sparse textual descriptions. Ideally, figures should be self-contained, where the caption can describe the figure in full detail.
3. While I appreciate the motivation in the introduction to learn long-horizon tasks, I do not think that the experiments necessarily study long-horizon tasks. The metaworld environments entail relatively short sequences of simple primitives, and furthermore this paper only uses 3 sub-skills which is not reflective of the diversity that comes with long-horizon tasks.
4. The writing currently suffers from a significant number of grammatical issues. There are a number of passive sentences throughout, and some sentences are simply difficult to follow, such as the following: “there is no enough ability in sub-skills to quickly adapt to new tasks, thus no certain sub-skill exceeds others to give out dominant supervision for the high-level network.” Other sentences such as “DMIL is a brand new iterative hierarchical meta-learning procedure” use informal language (“brand new”) and should be revised accordingly.
5. The discussion section mentions future avenues of research, but does not quite touch upon the core limitations of the work. Here are a set of potential limitations to warrant discussion: what are the failure modes encountered during experiments? was there mode collapse among the skills? how sensitive is the method to the number of skills K?
6. I’d like to know more about the MIL baseline — why was a transformer used here as opposed to a simple MLP, and why does DMIL not use a transformer?


**Summary Of The Paper:**

This paper proposes a meta imitation learning framework aimed at learning long-horizon robot control tasks with fast adaptation capabilities. Specifically, the proposed approach adapts the model-agnostic meta learning framework for learning a hierarchical policy, where both levels of the hierarchy are meta-trained and fine-tuned at test time to learn new tasks. On the metaworld benchmark, the proposed method outperforms prior work, and qualitative analysis shows that the method discovers meaningful skills.

**Summary Of The Review:**

While I think the technical contributions of the paper are sound and the experiments are generally thorough, this paper at its current form is limited by the writing quality. A signifiant amount of work needs to go in re-writing the introduction, improving the figures, and fixing grammatical issues.

---

> ### Author Response · Authors · 2021-11-21
> **Response to Reviewer ouLD (3/3)**
>
> **Q4: The writing currently suffers from a significant number of grammatical issues.**
>
> A4: Thanks for your suggestions. We have modified a number of grammatical issues in our original paper. Please check out our newly uploaded PDF.
>
> For the questions that you asked, *"there is no enough ability in sub-skills to quickly adapt to new tasks, thus no certain sub-skill exceeds others to give out dominant supervision for the high-level network."* means that the high-level network of DMIL-High can adapt to the new tasks, but no sub-skill can dominate for a long time period along the task like in DMIL in the original figure 5 since all sub-skills lack the quickly adaptation ability, which means they do not show good semantic characteristics like in DMIL.
>
> We have revised some informal languages and sentences as you pointed.
>
> **Q5: The discussion section does not quite touch upon the core limitations of the work. Here are a set of potential limitations to warrant discussion: what are the failure modes encountered during experiments? was there mode collapse among the skills? how sensitive is the method to the number of skills K?**
>
> A5: Thanks for your kind suggestions. We have modified the Discussion Section and upload the new PDF file. Please check out our newly uploaded PDF.
>
> We point two main limitations of DMIL in this section. Firstly, DMIL models all manipulation tasks as bi-level structures. However, in real-world situations, they may be multi-level structures. One can extend DMIL to multi-level hierarchical structures like done in recent works [3]. Secondly, DMIL does not capture temporal information in demonstrations. Future state conditioning in [4] seems an effective tool to improve few-shot imitation learning performance in long-horizon tasks such as in the Kitchen environments. Future works can employ temporal module such as transformer [1] as the high-level network of DMIL to improve its performance.
>
> For the three questions you asked, we answer them here separately:
>
> For the failure mode, DMIL mostly fails in relatively complex *picking* tasks such as *Shelf-place*. However, it is worth noticing that after the object slips from the robot hand, the robot can try to re-grasp it from current position (which is not shown in demonstrations). This shows the power of the strong adaptation ability of our  meta-learned hierarchical structure that can perform in unseen situations.
>
> For the mode collapse during the training process, we didn't encounter this situation in our experiments, i.e., every sub-skill gets well-trained even when $K=8$ in kitchen environment or $K=10$ in Meta-world environments. This may come from that, in the meta-training stage, adding more sub-skills can help the whole structure get lower loss, thus DMIL will use all of them for training. However, in our supplementary videos, we can see sub-skills trained with a large $K$ (for instance, $K=10$ in Meta-world environments) are not as semantic as sub-skills trained by a small $K$ (for instance, $K=3$ in Meta-world environments) during the execution of a task.
>
> We have discussed the problem about the number of skills K in A3.
>
> **Q6: I’d like to know more about the MIL baseline — why was a transformer used here as opposed to a simple MLP, and why does DMIL not use a transformer?**
>
> A6: We are sorry that our original paper may have confused you. MIL refers to [1], which is similar to the original one-shot imitation learning [2] but replaces the soft-attention network with a strong self-attention module (transformer). The reason why they use transformer may be that it is the state-of-the-art module that captures the temporal information and features in time-series data. We choose MIL as a baseline because it also uses Meta-world benchmark environments and perform few-shot imitation learning experiments in their paper [1], and the results can be a comparison for the effect of hierarchical structure.
>
> The reason why we didn't use transformer in DMIL is that unfortunately we don't have enough GPU memory to support the training of a hierarchical transformer-based deep network, and meta-learning this network needs even more memory since MAML needs to store intermediate parameters of transformer for its meta-updating. The transformer-based DMIL can be an interesting topic for future works.
>
> ----------
>
> [1] Cachet T, Perez J, Kim S. Transformer-based meta-Imitation learning for robotic manipulation[C]. Neural Information Processing Systems, Workshop on Robot Learning. 2020.
>
> [2] Duan Y, Andrychowicz M, Stadie B C, et al. One-shot imitation learning[J]. arXiv preprint arXiv:1703.07326, 2017.
>
> [3] Shu T, Xiong C, Socher R. Hierarchical and interpretable skill acquisition in multi-task reinforcement learning[J]. arXiv preprint arXiv:1712.07294, 2017.
>
> [4] Hakhamaneshi K, Zhao R, Zhan A, et al. Hierarchical few-shot imitation with skill transition models[J]. arXiv preprint arXiv:2107.08981, 2021.

---

> > ### Comment · Reviewer_ouLD · 2021-11-22
> > **Reviewer ouLD response to author rebuttal**
> >
> > I would like to thank the authors for the thorough rebuttal response. Several of my concerns on the writing quality have been addressed, and I am raising my recommendation to a 6. I'm still holding out on an 8, because figures 1 and 2 are still a bit busy and difficult to understand, and the experiments are not consolidated. For future versions of the manuscript I highly recommend that the authors evaluate all baselines (OptionGAIL, MIL, PEMIRL, MLSH, FIST, SPiRL) on all environments (metaworld and kitchen). Also the description on SPiRL needs more details -- is this baseline using an active RL exploration phase with skills (as described in the SPiRL paper), or is this a modified variant of the algorithm that operates on offline data only?

---

> > > ### Author Response · Authors · 2021-11-25
> > > **Response to the Response of Reviewer ouLD**
> > >
> > > Thanks for your appreciation of our work. We are sorry that we are still not able to give you a clear understanding of figures 1 and 2, and due to the short time of rebuttal, we are not able to perform consolidated experiments as you suggested. We will improve these two aspects in future versions.
> > >
> > > For the SPiRL problem, to adapt SPiRL to imitation learning, after pre-training the skills module, we fine-tune it on the downstream demonstrations (instead of finetuning with RL as proposed in the original paper). This is the same way done in [1].
> > >
> > > ------
> > >
> > > [1] Hakhamaneshi K, Zhao R, Zhan A, et al. Hierarchical few-shot imitation with skill transition models[J]. arXiv preprint arXiv:2107.08981, 2021.

---

> ### Author Response · Authors · 2021-11-21
> **Response to Reviewer ouLD (2/3)**
>
> **Q3: I do not think that the experiments necessarily study long-horizon tasks, and furthermore this paper only uses 3 sub-skills which is not reflective of the diversity that comes with long-horizon tasks.**
>
> A3: We add additional experiments in Kitchen experiments like done in [1]. We believe tasks in this environment are long-horizon. Results are as follows:
>
> 1. We perform DMIL on the Kitchen environment as the way in [2], and the table below shows the rewards of different methods on four unseen tasks in the Kitchen environment. FIST-no-FT refers to a variant of FIST [2] that does not use future conditioning, which makes the comparison fairer. DMIL achieves higher rewards on two out of four tasks and comparable results on the other two tasks, which exhibits the effectiveness of the bi-level meta-training procedure. The poor performance of DMIL on the first task may come from the choice of skill number $ K $ or from low-quality demonstrations. We perform ablation studies on $ K $ in the next table.
>
> | Task (Unseen) |  FIST-no-FT | SPiRL       | DMIL(ours) |
> |---|---|---|---|
> | Microwave, Kettle, **Top Burner**, Light Switch           | 2.0 ± 0.0   | **2.1 ± 0.48**  | 1.5±0.48   |
> | **Microwave**, Bottom Burner, Light Switch, Slide Cabinet | 0.0 ± 0.0   |  2.3 ± 0.49 | **2.35±0.39**  |
> | Microwave, **Kettle**, Hinge Cabinet, Slide Cabinet       | 1.0 ± 0.0   | 1.9 ± 0.29  | **3.15±0.22**  |
> | Microwave, Kettle, Hinge Cabinet, **Slide Cabinet**       | 2.0 ± 0.0   | **3.3 ± 0.38**  | 2.95±0.44  |
>
> 2. We perform ablation studies on skill number $ K $ in both Meta-world benchmarks and the kitchen environments, and results are as follows. In the Meta-world environments, we can see that a larger $ K $ can lead to higher success rates on meta-training tasks, but a smaller $ K $ can lead to better results on meta-testing tasks. This tells us that an excessive number of sub-skills can result in over-fitting of training data, and a smaller $ K $ can play the role of regularization. In the Kitchen environments, a larger number of skills can not lead to better testing results in unseen tasks.
>
> |  | ML10 | ML10 |ML10  |ML10  | ML45 |ML45  | ML45 | ML45 |
> |---|---|---|---|---|---|---|---|---|
> |  | Meta-training | Meta-training | Meta-testing | Meta-testing | Meta-training | Meta-training | Meta-testing | Meta-testing |
> | K | 1-shot | 3-shot | 1-shot | 3-shot | 1-shot | 3-shot | 1-shot | 3-shot |
> | 2 | 0.76 | 0.955 | 0.32 | **0.72** | 0.563 | 0.818 | **0.44** | **0.67** |
> | 3 | 0.775 | 0.949 | 0.396 | 0.71 | 0.59 | 0.859 | 0.376 | 0.64 |
> | 5 | 0.795 | 0.94 | **0.52** | 0.57 | 0.713 | 0.92 | 0.21 | 0.48 |
> | 10 | **0.8** | **0.975** | 0.38 | 0.62 | **0.736** | **0.931** | 0.34 | 0.64 |
>
>
>
> | Task (Unseen) |  K=2 | K=4 | K=8 |
> |---|---|---|---|
> | Microwave, Kettle, **Top Burner**, Light Switch | **1.9±0.43** | 1.5±0.48 | 1.7±0.22 |
> | **Microwave**, Bottom Burner, Light Switch, Slide Cabinet | 2.15±0.19 | **2.35±0.39** | 2.0±0.37 |
> | Microwave, **Kettle**, Hinge Cabinet, Slide Cabinet | 2.45±0.25 | **3.15±0.22** | 1.85±0.23 |
> | Microwave, Kettle, Hinge Cabinet, **Slide Cabinet** | 2.01±0.24 | **2.95±0.44** | 2.44±0.47 |
>
>
> We recommend you to check video results in supplementary materials.

---

> ### Author Response · Authors · 2021-11-21
> **Response to Reviewer ouLD (1/3)**
>
> Dear reviewer ouLD,
>
> We sincerely appreciate your valuable and insightful comments. We found them extremely helpful for improving our manuscript. We address each comment in detail, one by one below. We recommend you to check our new PDF for more details.
>
> **Q1: The introduction of the paper sells the work short. Little discussion is centered on “why” this proposed research problem is important, “why” previous approaches falls short, and “why” addressing the limitations of prior approaches is challenging.**
>
> A1: Thanks for your suggestions. We have modified the introduction of our paper. Please check out our newly uploaded PDF.
>
> In this revision, we add a representative example in figure 1 to better show why the research problem is important and why previous approaches fall short. We then modify the original fourth paragraph of the introduction to better emphasize why addressing the limitations of prior approaches is challenging. That is, designing a self-supervised hierarchical MAML algorithm is a novel problem, and the most challenging part is how to ensure convergence. To this question, in DMIL we design an elaborate training procedure that first fine-tunes the high-level network and sub-skills in sequence at inner loops, then meta-updates them simultaneously at outer loops. Furthermore, we theoretically prove the convergence of this special training procedure in our work.
>
> **Q2: The figures are difficult to follow. In particular, figure 1 and 2 have a lot of visual details but are accompanied with sparse textual descriptions. Ideally, figures should be self-contained, where the caption can describe the figure in full detail.**
>
> A2: Thanks for your suggestions. We have added more descriptions about original figures 1 and 2 in their captions, which explains figures in more detail. Please check out our newly uploaded PDF.

---

### Official Review · Reviewer_2AqY · 2021-11-02

**Correctness:** 4
**Technical Novelty And Significance:** 3
**Empirical Novelty And Significance:** 2
**Recommendation:** 6
**Confidence:** 2

**Main Review:**

**Pros**
- The authors show that DMIL is competitive against a thorough set of ablations on both ML10 and ML45 settings from meta-world.

- The convergence analysis hints at the algorithms training stability.

**Cons**
- The method mostly borrows components from existing works without explaining how their setting adds more complexity to the approach.

- From the results, it's difficult to compare this method against hierarchical imitation learning approaches that they cite (Yu et al., 2018b; Finn et al., 2017b; Yu et al., 2018a)

- The authors do not show that the results are robust across different tasks/environments (e.g., "Hierarchical Few-Shot Imitation with Skill Transition Models" looks at maze and kitchen environments) or number of subskills. I believe this is important because the manipulation tasks in meta-world may have compositional task structure that can't be assumed in other environments or tasks (such as navigation).

**Questions**
- How would you contrast DMIL against "Hierarchical Few-Shot Imitation with Skill Transition Models"?

**Feedback**
- It would aid understanding to include a algorithm box that outlines the order of inner and outer updates in one place.

**Summary Of The Paper:**

The authors propose to learn a high level policy network that picks a subskill policy to predict actions with meta imitation learning (DMIL). Both the high and low level policy networks are fine-tuned at meta-test time. They show that DMIL converges and evaluate their approach against several ablations on the ML10 and ML45 setting of the meta-world benchmark.

**Summary Of The Review:**

I vote to marginally reject this submission because the experimental results are not thorough enough to demonstrate robustness across environments and number of subskills and the novelty of DMIL mostly comes from combining known approaches. If more environments and subskills could be tested, I think the paper could be a candidate for submission.

---

> ### Author Response · Authors · 2021-11-21
> **Response to Reviewer 2AqY (3/3)**
>
> **Q4: How would you contrast DMIL against [4]?**
>
> A4: Besides the experiment result comparisons in A3, we would like to state one additional comparison here:
>
> Although [4] claims that their method can achieve few-shot imitation learning ability in new tasks, they just use fine-tune to realize this, and never use any other techniques such as MAML in our paper. As shown in [6], in most cases employing a meta-learning procedure can achieve better few-shot adaptation ability in new tasks than just fine-tuning. Thus our method is fundamentally more suitable for few-shot imitation learning tasks than [4]. It is worth mentioning that some context-based meta-learning methods such as [7] also aim to find a latent code to further condition the downstream network, which is similar to [4]. However, these methods never fine-tune the high-level network in new tasks like done in [4], which means that [4] cannot be seen as such kind of a meta-learning method.
>
>
> **Q5: It would aid understanding to include a algorithm box that outlines the order of inner and outer updates in one place.**
>
> A5: We already have an algorithm box at the section Appendix A in our original paper.
>
>
> ----------
>
>
> [1] Yu T, Abbeel P, Levine S, et al. One-shot hierarchical imitation learning of compound visuomotor tasks[J]. arXiv preprint arXiv:1810.11043, 2018.
>
> [2] Finn, Chelsea, et al. "One-shot visual imitation learning via meta-learning." Conference on Robot Learning. PMLR, 2017.
>
> [3] Yu T, Finn C, Xie A, et al. One-shot imitation from observing humans via domain-adaptive meta-learning[J]. arXiv preprint arXiv:1802.01557, 2018.
>
> [4] Hakhamaneshi K, Zhao R, Zhan A, et al. Hierarchical few-shot imitation with skill transition models[J]. arXiv preprint arXiv:2107.08981, 2021.
>
> [5] Duan Y, Andrychowicz M, Stadie B C, et al. One-shot imitation learning[J]. arXiv preprint arXiv:1703.07326, 2017.
>
> [6] Finn C, Abbeel P, Levine S. Model-agnostic meta-learning for fast adaptation of deep networks[C]//International Conference on Machine Learning. PMLR, 2017: 1126-1135.
>
> [7] Rakelly K, Zhou A, Finn C, et al. Efficient off-policy meta-reinforcement learning via probabilistic context variables[C]//International conference on machine learning. PMLR, 2019: 5331-5340.

---

> > ### Comment · Reviewer_2AqY · 2021-11-24
> > **Reviewer 2AqY Response to Author**
> >
> > Thank you for your thorough followup. I found A2 and A3 especially helpful. Many of my concerns have been addressed, so I am raising my score to a 6.

---

> ### Author Response · Authors · 2021-11-21
> **Response to Reviewer 2AqY (2/3)**
>
> **Q3: The authors do not show that the results are robust across different tasks/environments (e.g., maze and kitchen environments) or number of subskills.**
>
> A3: Thanks for your criticism. We add additional experiments of DMIL on the Kitchen environment as used in [4]. Results are as follows.
>
> 1. We perform DMIL on the Kitchen environment as the way in [2], and the table below shows the rewards of different methods on four unseen tasks in the Kitchen environment. FIST-no-FT refers to a variant of FIST [2] that does not use future conditioning, which makes the comparison fairer. DMIL achieves higher rewards on two out of four tasks and comparable results on the other two tasks, which exhibits the effectiveness of the bi-level meta-training procedure. The poor performance of DMIL on the first task may come from the choice of skill number $ K $ or from low-quality demonstrations. We perform ablation studies on $ K $ in the next table.
>
> | Task (Unseen) |  FIST-no-FT | SPiRL       | DMIL(ours) |
> |---|---|---|---|
> | Microwave, Kettle, **Top Burner**, Light Switch           | 2.0 ± 0.0   | **2.1 ± 0.48**  | 1.5±0.48   |
> | **Microwave**, Bottom Burner, Light Switch, Slide Cabinet | 0.0 ± 0.0   |  2.3 ± 0.49 | **2.35±0.39**  |
> | Microwave, **Kettle**, Hinge Cabinet, Slide Cabinet       | 1.0 ± 0.0   | 1.9 ± 0.29  | **3.15±0.22**  |
> | Microwave, Kettle, Hinge Cabinet, **Slide Cabinet**       | 2.0 ± 0.0   | **3.3 ± 0.38**  | 2.95±0.44  |
>
> 2. We perform ablation studies on skill number $ K $ in both Meta-world benchmarks and the kitchen environments, and results are as follows. In the Meta-world environments, we can see that a larger $ K $ can lead to higher success rates on meta-training tasks, but a smaller $ K $ can lead to better results on meta-testing tasks. This tells us that an excessive number of sub-skills can result in over-fitting of training data, and a smaller $ K $ can play the role of regularization. In the Kitchen environments, a larger number of skills can not lead to better testing results in unseen tasks.
>
> |  | ML10 | ML10 |ML10  |ML10  | ML45 |ML45  | ML45 | ML45 |
> |---|---|---|---|---|---|---|---|---|
> |  | Meta-training | Meta-training | Meta-testing | Meta-testing | Meta-training | Meta-training | Meta-testing | Meta-testing |
> | K | 1-shot | 3-shot | 1-shot | 3-shot | 1-shot | 3-shot | 1-shot | 3-shot |
> | 2 | 0.76 | 0.955 | 0.32 | **0.72** | 0.563 | 0.818 | **0.44** | **0.67** |
> | 3 | 0.775 | 0.949 | 0.396 | 0.71 | 0.59 | 0.859 | 0.376 | 0.64 |
> | 5 | 0.795 | 0.94 | **0.52** | 0.57 | 0.713 | 0.92 | 0.21 | 0.48 |
> | 10 | **0.8** | **0.975** | 0.38 | 0.62 | **0.736** | **0.931** | 0.34 | 0.64 |
>
>
>
> | Task (Unseen) |  K=2 | K=4 | K=8 |
> |---|---|---|---|
> | Microwave, Kettle, **Top Burner**, Light Switch | **1.9±0.43** | 1.5±0.48 | 1.7±0.22 |
> | **Microwave**, Bottom Burner, Light Switch, Slide Cabinet | 2.15±0.19 | **2.35±0.39** | 2.0±0.37 |
> | Microwave, **Kettle**, Hinge Cabinet, Slide Cabinet | 2.45±0.25 | **3.15±0.22** | 1.85±0.23 |
> | Microwave, Kettle, Hinge Cabinet, **Slide Cabinet** | 2.01±0.24 | **2.95±0.44** | 2.44±0.47 |
>
>
> It is worth mentioning that some key differences between the Kitchen environment and Meta-world environments:
>
> Firstly, although there are 24 *different* tasks in the Kitchen environment, they are actually in the same environment, with identical observation space (which means the same object is represented in the same position in the state vector). *Difference tasks* means the different sequences to accomplish the five sub-tasks. Instead, in Meta-world, different tasks are in different environments, where the same position in the state vector may represent different objects (for example, the hammer in the *hammer-v2* task and the cube in the *push-v2* task are both at the same dimensions in the state vector). Obviously, manipulating different objects in different environments requires different skills, which makes Meta-world more challenging than the Kitchen environment.
>
> Secondly, the amounts of demonstrations of Kitchen environment and Meta-world environments are extremely different. There are 24 different tasks in the Kitchen environment, and in each task, there are about 10 to 30 demonstration trajectories. However, in our experiments, we collected 2000 demonstration trajectories for each task in Meta-world environments. This may show that the Kitchen environment is a much easier environment. As a reference, in most robot imitation learning tasks, we usually need to collect several thousands of demonstrations to get good few-shot imitation learning results, such as in [1], [2], and [5].

---

> ### Author Response · Authors · 2021-11-21
> **Response to Reviewer 2AqY (1/3)**
>
> Dear reviewer 2AqY,
>
> We sincerely appreciate your valuable and insightful comments. We found them extremely helpful for improving our manuscript. We address each comment in detail, one by one below. We recommend you to check our new PDF for more details.
>
> **Q1: The method mostly borrows components from existing works without explaining how their setting adds more complexity to the approach.**
>
> A1: DMIL incorporates MAML and HIL and builds a hierarchical imitation learning structure that meta-learns both levels. We have discussed the complexities and challenges caused by them in the introduction, which contains two aspects of challenges. Firstly, there is no reason that the supervision of HIL can be directly used for hierarchical meta-learning structure, thus we need to design appropriate forms of supervision for the meta-learning of each level. Secondly, MAML and HIL are both two-fold processes, thus DMIL becomes a generally four-fold process. How to schedule the inner meta-learning loops and outer meta-learning loops of two levels in HIL to ensure convergence is a problem. We successfully tackle these two difficulties in our work, as shown in the section of introduction and method. Concretely, we design a special training procedure that first fine-tunes the high-level network and sub-skills in sequence at inner loops, then meta-updates them simultaneously at outer loops. As pointed out by reviewer ouLD, *while the proposed approach integrates existing elements from MAML and hierarchical imitation learning, it does so in a unique and novel manner.*
>
> **Q2: It's difficult to compare DMIL against hierarchical imitation learning approaches such as [1], [2], and [3] from our results.**
>
> A2: We have discussed the differences between our problem settings and the settings in [1], [2], and [3] in the second paragraph of section 2.2.
>
> [1] uses several manually-designed pretraining tasks to pretrain a set of skill primitives such as *grasping*, *pushing* and *reaching*, and then use a hierarchical structure to train them, which is not the case in our setting where the provided demonstrations are unsegmented, thus we cannot design a set of pretrain tasks to get pretrained primitives. Furthermore, they didn't adapt the high-level network in new tasks, which is instead one of the main contributions of our work. Indeed, [1] is similar to DMIL-Low (one of the ablation variants in our experiments), which can be seen as firstly acquiring a set of primitives with unsupervised skill discovery method (EM here) and then learning a fixed high-level network over them, thus it can fine-tune sub-skills in new tasks with MAML. We have shown its bad performance in meta-testing tasks.
>
> [2] and [3] only learn one monolithic policy which is totally different from our setting. Thus in our original paper, we didn't conduct experiments to compare DMIL with them.

---

### Official Review · Reviewer_ZyiJ · 2021-11-03

**Correctness:** 4
**Technical Novelty And Significance:** 3
**Empirical Novelty And Significance:** 3
**Recommendation:** 8
**Confidence:** 4

**Main Review:**

The strength of the paper is that they proposed a natural extension of meta-IL and HIL. The total algorithm is reasonable, and its convergence is proved based on the theory of the EM algorithm. The connection to the EM algorithm is natural and understandable. The experiment showed the usefulness of the method as well.
The weakness of the paper is the limitation of evaluations. They tested their methods on two pre-existing simulation environments. The evaluation using a task meta and hierarchical nature is more explicitly required is expected. The strength of the method would be more clearly shown with such a task.


**Summary Of The Paper:**

The paper describes Dual Meta Imitation Learning (DMIL) which is a hierarchical meta imitation learning method where the high-level network and sub-skills are iteratively meta-learned with model-agnostic meta-learning (MAML). The DMIL is a hierarchical extension of MAML-based imitation learning (IL) and is a meta-learning extension of Hierarchical Imitation learning (HIL).  The authors provide theoretical proof of the convergence based on the connection with the Expectation-Maximization (EM) algorithm. They showed the state-of-the-art few-shot imitation learning performance on the meta-world benchmark.

**Summary Of The Review:**

The paper proposed a new meta HIL method called DMIL. The method is clearly explained. The background is described in a proper manner and is informative. The novelty is clear. Also, the empirical evaluation is appropriate though the number of environments can be increased.
As a whole, I admit that the paper is an excellent paper.

---

> ### Author Response · Authors · 2021-11-21
> **Response to Reviewer ZyiJ**
>
> Dear reviewer ZyiJ,
>
> We sincerely appreciate your valuable and insightful comments. We address your questions in detail below. We recommend you to check our new PDF for more details.
>
> **Q: The evaluation using a task meta and hierarchical nature is more explicitly required is expected.**
>
> A: We add additional experiments of our method on the Kitchen environment as used in [1] and [2], which is a more representative long-horizon environment in the hierarchical learning domain than Meta-World benchmark environments. The results are as follows. More results can be found in our paper.
>
> 1. We perform DMIL on the Kitchen environment as the way in [2], and the table below shows the rewards of different methods on four unseen tasks in the Kitchen environment. FIST-no-FT refers to a variant of FIST [2] that does not use future conditioning, which makes the comparison fairer. DMIL achieves higher rewards on two out of four tasks and comparable results on the other two tasks, which exhibits the effectiveness of the bi-level meta-training procedure. The poor performance of DMIL on the first task may come from the choice of skill number $ K $ or from low-quality demonstrations. We perform ablation studies on $ K $ in the next table.
>
> | Task (Unseen) |  FIST-no-FT | SPiRL       | DMIL(ours) |
> |---|---|---|---|
> | Microwave, Kettle, **Top Burner**, Light Switch           | 2.0 ± 0.0   | **2.1 ± 0.48**  | 1.5±0.48   |
> | **Microwave**, Bottom Burner, Light Switch, Slide Cabinet | 0.0 ± 0.0   |  2.3 ± 0.49 | **2.35±0.39**  |
> | Microwave, **Kettle**, Hinge Cabinet, Slide Cabinet       | 1.0 ± 0.0   | 1.9 ± 0.29  | **3.15±0.22**  |
> | Microwave, Kettle, Hinge Cabinet, **Slide Cabinet**       | 2.0 ± 0.0   | **3.3 ± 0.38**  | 2.95±0.44  |
>
> 2. We perform ablation studies on skill number $ K $ in both Meta-world benchmarks and the kitchen environments, and results are as follows. In the Meta-world environments, we can see that a larger $ K $ can lead to higher success rates on meta-training tasks, but a smaller $ K $ can lead to better results on meta-testing tasks. This tells us that an excessive number of sub-skills may result in over-fitting on training data, and a smaller $ K $ can play the role of regularization. In the Kitchen environments, a larger number of skills can not lead to better testing results in unseen tasks.
>
> |  | ML10 | ML10 |ML10  |ML10  | ML45 |ML45  | ML45 | ML45 |
> |---|---|---|---|---|---|---|---|---|
> |  | Meta-training | Meta-training | Meta-testing | Meta-testing | Meta-training | Meta-training | Meta-testing | Meta-testing |
> | K | 1-shot | 3-shot | 1-shot | 3-shot | 1-shot | 3-shot | 1-shot | 3-shot |
> | 2 | 0.76 | 0.955 | 0.32 | **0.72** | 0.563 | 0.818 | **0.44** | **0.67** |
> | 3 | 0.775 | 0.949 | 0.396 | 0.71 | 0.59 | 0.859 | 0.376 | 0.64 |
> | 5 | 0.795 | 0.94 | **0.52** | 0.57 | 0.713 | 0.92 | 0.21 | 0.48 |
> | 10 | **0.8** | **0.975** | 0.38 | 0.62 | **0.736** | **0.931** | 0.34 | 0.64 |
>
>
>
> | Task (Unseen) |  K=2 | K=4 | K=8 |
> |---|---|---|---|
> | Microwave, Kettle, **Top Burner**, Light Switch | **1.9±0.43** | 1.5±0.48 | 1.7±0.22 |
> | **Microwave**, Bottom Burner, Light Switch, Slide Cabinet | 2.15±0.19 | **2.35±0.39** | 2.0±0.37 |
> | Microwave, **Kettle**, Hinge Cabinet, Slide Cabinet | 2.45±0.25 | **3.15±0.22** | 1.85±0.23 |
> | Microwave, Kettle, Hinge Cabinet, **Slide Cabinet** | 2.01±0.24 | **2.95±0.44** | 2.44±0.47 |
>
>
> ----------
>
>
> [1] Pertsch K, Lee Y, Lim J J. Accelerating reinforcement learning with learned skill priors[J]. arXiv preprint arXiv:2010.11944, 2020.
>
> [2] Hakhamaneshi K, Zhao R, Zhan A, et al. Hierarchical few-shot imitation with skill transition models[J]. arXiv preprint arXiv:2107.08981, 2021.

---

### Author Response · Authors · 2021-11-21
**General response: Paper revision**

We thank all reviewers for their valuable comments and constructive suggestions. Accordingly, we have revised the paper and conducted additional experiments motivated by several comments. We summarize the major improvements of the paper during revision as follows:

1. We modified the introduction of the original paper. Concretely, we add a representative example in figure 1 to better introduce “why” the proposed research problem is important and “why” addressing the limitations of prior approaches is challenging, as reviewer ouLD required.

2. We have conducted extensive additional experiments on the Kitchen environments and ablation studies about different skill number K, the bi-level meta-learning process, different fine-tuning time and comparison about soft/hard EM algorithms in DMIL. We add these results to the Experiment section and appendix.

3. We have provided more clear video results in supplementary materials.

4. Presentation improvement: We modified or simplified some expressions in the original paper to improve clarity, and corrected typos in the paper.

---

### Decision · Program_Chairs · 2022-01-20

**Decision:**

Reject

**Comment:**

The paper proposes a hierarchical meta imitation learning framework for few-shot transfer in the context of long-horizon control tasks. Underlying the framework is a hierarchical adaptation of model-agnostic meta learning (MAML) that jointly learns the high-level policy together with the set of modular low-level policies (sub-skills), both of which are fine-tuned at test time based on a small number of demonstrations. Experimental evaluations on the meta-world benchmark as well as a kitchen environment benchmark compare the proposed framework with recent baselines.

As several reviewers note, the problem of jointly learning modular policies together with the high-level policy for composing these sub-skills is both challenging and interesting to the robotics and learning communities. The manner by which the paper extends existing work in meta-learning (MAML) and hierarchical imitation learning is novel and technically sound. The reviewers raised some concerns, notably those regarding (1) the the framework's sensitivity to various hyperparameter settings and its ability to generalize to other domains; (2) the merits of joint optimization over decoupled optimization of the sub-skills and high-level policy; and (3) the need for experiments/evaluations on different domains. The authors provided a detailed response to each of the reviewers that includes the addition of a different benchmark evaluation (the kitchen environment), new ablation studies, and updates to the text. After a thorough review, however, concerns remain regarding the reproducibility of the results, which call into question some of the key contributions that the paper claims to provide over the existing state-of-the-art. The authors are encouraged to provide a more balanced discussion of the contributions along with evidence to support reproducibility in any future version of the paper.

---

> ### Public Comment · ~Chongkai_Gao1 · 2022-02-22
> **Final Comments**
>
> We appreciate the PC's comments and the effort of the AC and reviewers in giving us valuable feedback. Here we would like to add a final comment for the general public.
>
> 1. Our paper received final scores of 6, 6, 6, 8, which represents the recognition of our work by reviewers.
>
> 2. We are respectfully opposed to the way in which Program Chairs questioned our paper during the rebuttal. During the rebuttal period, we received frequent questions from Program Chairs in **email** form about the reproducibility of our paper. Since ICLR is a conference based on the OpenReview platform, we think these discussions should take place here in this OpenReview platform, rather than in emails.
>
> 3. We respectfully disagree with the PC's comment regarding the reproducibility of our paper. We didn't get further questions or feedback about our paper after we give our source code and trained models at https://anonymous.4open.science/r/DMIL to Program Chairs. We will provide a more balanced discussion of the contributions along with evidence to support the reproducibility of our paper.
>
> We hope that the community finds our work useful and builds upon it.